# USP7 controls NGN3 stability and pancreatic endocrine lineage development

Teodora Manea [1,7], Jessica Kristine Nelson[2,3,7], Cristina Maria Garrone [1], Karin Hansson [3], Ian Evans [2,3], Axel Behrens [2,3,4,5] & Rocio Sancho [1,6] ✉

Understanding the factors and mechanisms involved in beta-cell development will guide therapeutic efforts to generate fully functional beta cells for diabetes. Neurogenin 3 (NGN3) is the key transcription factor that marks endocrine progenitors and drives beta-cell differentiation. Here we screen for binding partners of NGN3 and identify the deubiquitylating enzyme USP7 as a key regulator of NGN3 stability. Mechanistically, USP7 interacts with, deubiquitinates and stabilizes NGN3. In vivo, conditional knockout of *Usp7* in the mouse embryonic pancreas causes a dramatic reduction in islet formation and hyperglycemia in adult mice, due to impaired NGN3-mediated endocrine specification during pancreatic development. Furthermore, pharmacological inhibition of USP7 during endocrine specification in human iPSC models of beta-cell differentiation decreases NGN3 expressing progenitor cell numbers and impairs beta cell differentiation. Thus, the USP7-NGN3 axis is an essential mechanism for driving endocrine development and beta-cell differentiation, which can be therapeutically exploited.

The generation of differentiated cell lineages during embryonic pancreas organogenesis requires precisely coordinated cell fate choices[1]. These cell fate decisions culminate in the formation of the exocrine (ductal and acinar cells) and endocrine (beta, alpha, delta, pancreatic polypeptide, and epsilon cells) compartments in the adult pancreas where they exert digestive and glucose homeostatic functions, respectively[2–7]. PDX1-expressing (PDX1+) pancreatic progenitors, first observed at embryonic day (E) 8.5 of mouse development[8] and at week (W) 4 of human development[9], give rise to both exocrine and endocrine cells upon the orchestrated action of specific transcription factors. The basic helix-loop-helix (bHLH) transcription factor Neurogenin 3 (NGN3), encoded by the *NEUROG3* gene, plays an essential role in endocrine specification[10,11], with its loss in the mouse pancreas leading to complete ablation of endocrine cells[12]. Biphasic expression of NGN3 in the murine pancreas, with a first wave between E10.5 and E11.5[8] and a second wave between E12 and E18.5[8,13], peaking at E15.5[8,14] is required for correct beta-cell formation[12]. NGN3 expression during human pancreatic development differs from murine development, with NGN3 peaking between W10–14, followed by a sharp decrease by W18–21[9,15]. Despite these differences in NGN3 expression patterns between mouse and human, transient NGN3 expression during pancreatic development is vital for endocrine specification in both species[11,12,16], and the precise regulation of NGN3 expression peaks at specific times during embryogenesis is essential for beta-cell formation[17]. However, despite the crucial role of NGN3 in beta-cell differentiation, the fundamental NGN3 regulatory network is not yet fully understood.

While NGN3 regulation has been mainly studied at the transcriptional level[18–23], it has been suggested that post-translational mechanisms are responsible for fine-tuning NGN3 levels during the crucial cell

---

[1]Centre for Gene Therapy and Regenerative Medicine, King's College London, London, UK. [2]Adult Stem Cell Laboratory, The Francis Crick Institute, 1 Midland Road, London NW1 1AT, UK. [3]Cancer Stem Cell Laboratory, The Breast Cancer Now Toby Robins Research Centre, Institute of Cancer Research, 237 Fulham Road, London SW3 6JB, UK. [4]Imperial College, Division of Cancer, Department of Surgery and Cancer, Imperial College, Exhibition Road, London SW7 2AZ, UK. [5]Convergence Science Centre, Imperial College, Exhibition Road, London SW7 2BU, UK. [6]Department of Internal Medicine III, University Hospital Carl Gustav Carus at the Technische Universität Dresden, Dresden, Germany. [7]These authors contributed equally: Teodora Manea, Jessica Kristine Nelson. ✉e-mail: rocio.sancho@kcl.ac.uk

fate decision of PDX1+ progenitors to become endocrine progenitors[24,25]. NGN3 is tightly regulated post-translationally, with its phosphorylation levels playing a role in its degradation through the ubiquitin-proteasome system[25,26]. Specifically, phosphorylation at the S183 site of the mouse NGN3 protein facilitates its interaction with E3 ubiquitin ligase FBW7, followed by NGN3 ubiquitination and degradation[24]. However, while FBW7 loss induces ductal-to-endocrine plasticity via NGN3 stabilization, the effect is relatively modest[24], suggesting that additional pathways may be involved in NGN3 regulation during pancreatic development.

In this study, we identify ubiquitin-specific peptidase 7 (USP7) as the first known deubiquitinase to interact with NGN3 and regulate its stability in both in vitro and in vivo models of pancreatic development. USP7 overexpression leads to deubiquitination of exogenous NGN3, resulting in a significant increase in NGN3 stability due to decreased proteasomal degradation. Furthermore, Pdx1-conditional knockout of *Usp7* in the mouse embryonic pancreas leads to significantly decreased numbers of NGN3-expressing (NGN3+) endocrine progenitors at E14.5 and almost complete ablation of the endocrine compartment. Finally, we show that USP7 inhibition during human iPSC-to-beta-cell differentiation results in a drastic reduction in NGN3+ endocrine progenitors and insulin-expressing (INS+) beta-like cells. Our study demonstrates the crucial role USP7 plays in mouse and human endocrine specification by deubiquitinating and stabilizing NGN3 and highlights the importance of the USP7-NGN3 axis in the generation of beta cells.

## Results

### USP7 deubiquitinates and stabilizes NGN3 in vitro

Given the important role of NGN3 in pancreatic development, we set out to identify regulators of NGN3 using a systematic, unbiased approach in which immunoprecipitation of an ectopically expressed N-terminally HA-tagged NGN3 in HEK293A cells was followed by mass spectrometry (IP-MS; Supplementary Fig. 1a). This approach led to the identification of 49 interactors (Supplementary Fig. 1b, c) that are involved in various biological pathways (Supplementary Fig. 1d). Amongst the top 10 biological processes, several candidate interactors with known ubiquitin and ubiquitin-like protein ligase binding functions were identified, including the deubiquitinase USP7 (Supplementary Fig. 1c). Subsequent co-immunoprecipitation experiments in HEK293A cells showed that NGN3 directly interacted with endogenous USP7 (Supplementary Fig. 1e), ectopically expressed Flag-tagged wild-type USP7 (Flag-USP7$^{WT}$), and catalytically inactive USP7 (Flag-USP7$^{C223A}$) (Fig. 1a–c). We further investigated the biological effect of this interaction by overexpressing tagged versions of NGN3 and USP7 in HEK293A cells (Fig. 1a), and observed that only USP7$^{WT}$ was able to stabilize NGN3 in a dose-dependent manner, whereas the catalytically inactive mutant USP7$^{C223A}$ had no effect on NGN3 protein levels (Fig. 1d, e and Supplementary Fig. 2a). Due to the role of USP7 in substrate deubiquitination[27,28], we next hypothesized that USP7-mediated stabilization of NGN3 occurred post-translationally. To test this, we performed cycloheximide chase experiments, in which protein synthesis is inhibited. While overexpression of Flag-USP7$^{WT}$ was able to fully rescue the limited half-life (-15 min) of ectopically expressed NGN3, NGN3 half-life was not affected upon Flag-USP7$^{C223A}$ overexpression (Fig. 1f, g). These data suggested that the catalytic activity of USP7 is required for NGN3 post-translational stabilization. To address whether USP7 overexpression affected NGN3 ubiquitination, we carried out NGN3 ubiquitination assays. We used Tandem Ubiquitin Binding Entities (TUBEs) pull downs to enrich for ubiquitinated NGN3 in lysates from HEK293A cells transfected with HA-NGN3, His-tagged ubiquitin, and either Flag-USP7$^{WT}$ or Flag-USP7$^{C223A}$. As ubiquitination is known to play a role in proteasomal degradation, we treated all cells with the proteasomal inhibitor MG132, to prevent loss of ubiquitinated NGN3 protein prior to analysis. NGN3 showed a clear ubiquitination ladder in the absence of Flag-USP7$^{WT}$ (Fig. 2a–d). Importantly, Flag-

USP7$^{WT}$ overexpression significantly reduced NGN3 ubiquitination levels in a USP7 dose-dependent manner (Fig. 2a, b), while no reduction of NGN3 ubiquitination was observed when overexpressing the catalytically inactive Flag-USP7$^{C223A}$ (Fig. 2c, d). Consistent with previous reports demonstrating that NGN3 is degraded through the ubiquitin-proteasome system[26,29], findings from our cycloheximide chase and ubiquitination experiments (Figs. 1f, g, 2a–d) indicate that USP7-dependent stabilization of NGN3 results from reduced NGN3 polyubiquitination and proteasomal degradation.

The specific linkage between ubiquitin molecules is important in regulating the effect of a ubiquitination event on the substrate. While K48-linked ubiquitination targets proteins for proteasomal degradation, K63-linked ubiquitination is linked to other functions, including protein–protein interactions[30]. Therefore, we next investigated whether USP7 deubiquitinase activity was specific to K48 or K63 linkages by analyzing the levels of ubiquitinated NGN3 in HEK293A cells overexpressing HA-NGN3, Flag-USP7$^{WT}$ and His-tagged ubiquitin mutated at K48 or K63 (Ub$^{K48R}$-His or Ub$^{K63R}$-His) (Fig. 2e, f). Interestingly, USP7 overexpression significantly decreased the NGN3 polyubiquitination ladder in both Ub$^{K48R}$-His and Ub$^{K63R}$-His mutants suggesting that USP7 mediates the overall deubiquitination of NGN3 with potential consequences not only for NGN3 stability but also for its interactions with other proteins (Fig. 2e, f).

NGN3 phosphorylation has been previously shown to be required for its ubiquitination by the E3 ubiquitin ligase FBW7[24]. NGN3$^{6SA}$, which contains serine-to-alanine mutations at six predicted phosphorylation sites (S14, S38, S160, S174, S183, and S199), is significantly more stable when compared to wild-type NGN3 (NGN3$^{WT}$) due to low phosphorylation and impaired interaction with E3 ubiquitin ligases[25]. To determine whether NGN3 phosphorylation also plays a role in USP7-mediated stabilization, we tested the ability of the NGN3$^{6SA}$ phospho-mutant to interact with USP7. We found that the multi-site mutations in NGN3$^{6SA}$ significantly decrease its interaction with USP7, compared to NGN3$^{WT}$ (Fig. 2g, h). Moreover, NGN3$^{6SA}$ exhibits lower overall ubiquitination levels, which were insensitive to USP7 overexpression (Fig. 2i, j). In contrast to NGN3$^{6SA}$, single-site NGN3 phosphorylation mutants (Supplementary Fig. 2b, c) did not exhibit a decreased ability to pull down USP7, with most of them being stabilized by USP7 overexpression (Supplementary Fig. 2d, e). This indicates that the relatively reduced ubiquitination levels of NGN3$^{6SA}$, rather than the impairment of individual phosphorylation sites within the protein, lead to its decreased interaction with USP7. Collectively, these findings suggest that USP7 directly interacts with ubiquitinated NGN3, preventing its proteasomal degradation and increasing NGN3 protein stability.

### Loss of USP7 leads to reduced endocrine lineage specification and diabetes

Since NGN3 is a crucial pancreatic proendocrine factor, we next investigated the effect of *Usp7* loss on pancreatic homeostasis. To this end, we generated pancreas-specific conditional *Usp7* knockout mice. The *Usp7* conditional knockout was previously created by inserting loxP sites flanking exon 6 of the *Usp7* gene[31] to allow for the deletion of exon 6 upon pancreas-specific (*Pdx1* promoter-driven) expression of Cre recombinase. Deletion of exon 6 disrupts the catalytic domain of USP7, removing the conserved catalytic site (cysteine 223), and causes truncation of the rest of the protein due to a reading frame shift that ensures loss of USP7 function[31] (Fig. 3a). Interestingly, we found that mice with conditional homozygous deletion of *Usp7* (*Pdx1-Cre*; *Usp7*$^{F/F}$) were viable but displayed reduced body weight at 5–7 weeks post birth when compared to *Usp7* wild-type (*Usp7*$^{F/F}$) or heterozygous (*Pdx1-Cre*; *Usp7*$^{F/+}$) mice (Fig. 3b). *Pdx1-Cre*; *Usp7*$^{F/F}$ mice showed no significant difference in pancreas/total body weight ratio, yet developed hyperglycemia at 5–7 weeks of age (Fig. 3b). Moreover, while the acinar and ductal compartments in the pancreas appeared morphologically normal (Supplementary Fig. 3a), homozygous deletion of *Usp7* led to a

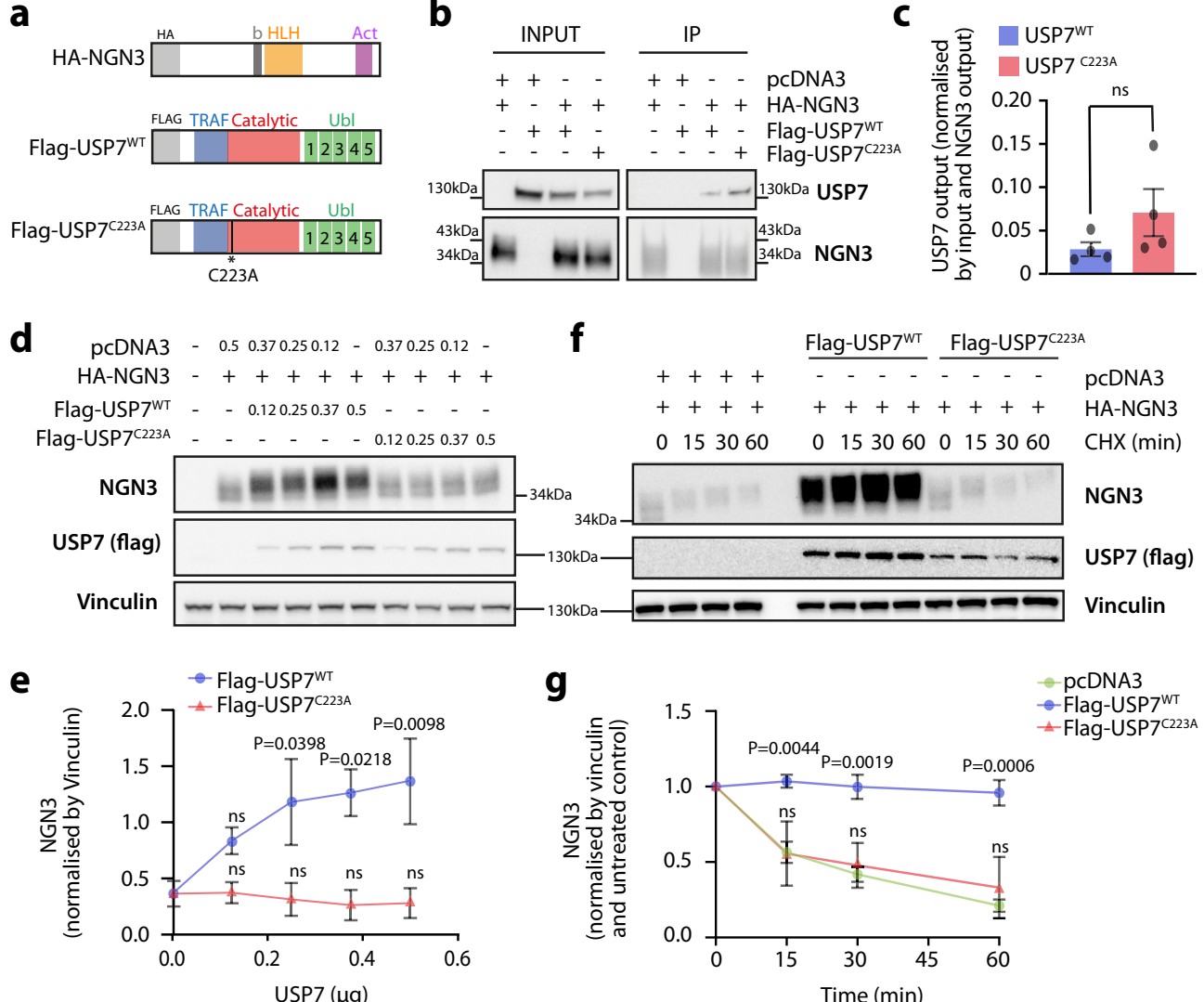

**Fig. 1 | USP7 modulates NGN3 stability through direct protein–protein inter-actions in HEK293A cells. a** Schematic of the protein structures of HA-NGN3, Flag-USP7[WT], and Flag-USP7[C223A]. **b** Immunoblotting for NGN3 and USP7 (Flag) in an HA-NGN3 immunoprecipitation experiment in HEK293A cells. **c** Quantification of Flag-USP7[WT] and Flag-USP7[C223A] in the output of HA-NGN3 immunoprecipitation experiments, normalized by immunoprecipitated NGN3 levels and USP7 input levels in each sample. Bar graphs represent mean ± SEM, and statistical significance was determined by two-tailed Student's *t*-test (*n* = 4 biologically independent experiments). **d** Immunoblotting for NGN3 in HEK293A samples co-transfected with increasing concentrations of Flag-USP7[WT] or Flag-USP7[C223A]. **e** Quantification of NGN3, normalized by Vinculin (*n* = 3 biologically independent experiments). The plot represents mean ± SEM, and statistical significance was determined by two-way

ANOVA with Dunnett's multiple comparison tests against the Control samples (HA-NGN3 + pcDNA3 empty vector). **f** Immunoblotting of NGN3 in HEK293A samples co-transfected with either Flag-USP7[WT], Flag-USP7[C223A], or empty pcDNA3 vector, after 0, 15, 30, or 60 min of cycloheximide treatment. **g** Quantification of NGN3 protein levels throughout cycloheximide chase experiments in samples co-transfected with either Flag-USP7[WT], Flag-USP7[C223A], or empty pcDNA3 vector, normalized by Vinculin and by the untreated control (0 min) for each condition (*n* = 4 biologically independent experiments). The plot represents mean ± SEM, and statistical significance was determined by two-way ANOVA with Dunnett's multiple comparison tests against pcDNA3 co-transfected sample from the corresponding timepoint. Source data are provided as a Source Data file.

dramatic reduction in islet number and size that was readily apparent in H&E staining of whole pancreas tissue sections (Supplementary Fig. 3a). Expression of USP7 in adult pancreas was detected in the islets of Langerhans and in scattered ductal cells, while acinar cells were negative for USP7 (Supplementary Fig. 3b). Consistent with an efficient deletion, we did not detect any USP7 signal in *Pdx1-Cre*; *Usp7*[F/F] adult pancreas (Supplementary Fig. 3b). Further immunohistochemistry analysis of the pancreas demonstrated a clear reduction in the number of INS+ islets of Langerhans in the *Pdx1-Cre*; *Usp7*[F/F] mice when com-pared to *Usp7*[F/F] control mice (Fig. 3c, d). To check whether loss of USP7 had any effect on the ductal and acinar compartment, we per-formed immunofluorescent staining (IF) for amylase (AMY; acinar cell marker), keratin 19 (CK19; ductal cell marker) and insulin (INS; beta cell

marker) in the pancreas of *Usp7*[F/F] and *Pdx1-Cre*; *Usp7*[F/F] mice (Fig. 3e). While no significant difference was detected in the % of amylase and CK19 area in the *Pdx1-Cre*; *Usp7*[F/F] when compared to Usp7[F/F] control mice, a severe decrease in the % of INS+ cells area in *Pdx1-Cre*; *Usp7*[F/F] mice was observed (Fig. 3f). The islets of Langerhans contain alpha, beta and delta cells that produce glucagon (GCG), c-peptide/insulin, and somatostatin (SST), respectively. Further analysis of the islets demonstrated that deletion of *Usp7* in the *Pdx1-Cre*; *Usp7*[F/F] mice resulted in reduced expression of c-peptide, GCG, and SST protein levels, compared to control mice (Fig. 3g, h and Supplementary Fig. 3c). Taken together, these findings suggest that USP7 plays a key role in pancreatic endocrine development and the ability to regulate circulating blood glucose levels in mice.

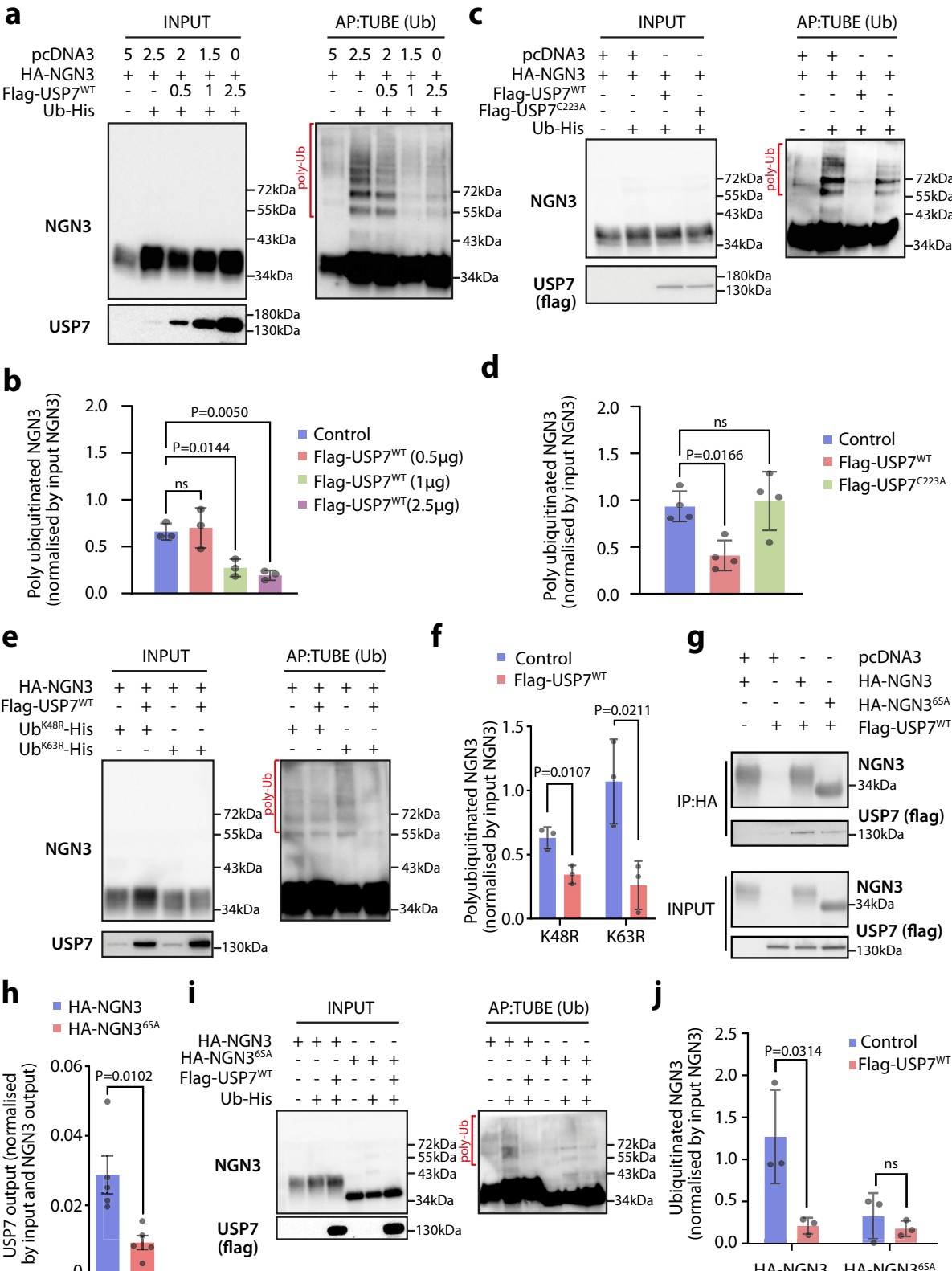

## USP7 regulates NGN3-mediated endocrine formation during development

During mouse pancreatic development, NGN3 expression is detected in two waves, the first between E10.5-E11.5 and the second between E12–E18.5, peaking at E15.5[8,13,32]. Since USP7 was identified as an interactor of NGN3 in our IP-MS experiments (Supplementary Fig. 1), and the loss of *Usp7* in mouse pancreas led to a severe reduction in the adult

pancreatic endocrine compartment (Fig. 3), we reasoned that the phenotype observed in *Pdx1-Cre*; *Usp7^F/F* mice could be due to defective pancreatic development during the NGN3 expression window. To test this, we generated *Usp7^F/F* and *Pdx1-Cre*; *Usp7^F/F* E12.5 and E14.5 mouse embryos (Fig. 4a and Supplementary Fig. 4a) and analyzed broad pancreatic morphology, as well as expression of NGN3 and other downstream endocrine-specific markers. As expected, H&E staining revealed

**Fig. 2 | USP7 overexpression leads to NGN3 deubiquitination in HEK293A cells.**
**a** Immunoblotting for exogenous NGN3 in tandem ubiquitin-binding entities (TUBE) affinity purification (AP) assays in MG132-treated HEK293A cells, with or without USP7 co-transfection at indicated concentrations (μg). **b** Quantification of polyubiquitinated NGN3 co-transfected with increasing concentrations of Flag-USP7[WT], normalized by input NGN3. The bar graph represents mean ± SEM, and statistical significance was determined by one-way ANOVA with the Dunnett post hoc test (*n* = 3 biologically independent experiments). **c** Immunoblotting for exogenous NGN3 in TUBE affinity purification assays in MG132-treated HEK293A cells after co-transfection with 2.5 μg Flag-USP7[WT], Flag-USP7[C223A], or empty pcDNA3 control. **d** Quantification of polyubiquitinated NGN3, normalized by input NGN3. The bar graph represents mean ± SEM, and statistical significance was determined by one-way ANOVA with the Dunnett post hoc test (n = 4 biologically independent experiments). **e** Immunoblotting for exogenous NGN3 in TUBE affinity purification assays in MG132-treated HEK293A cells with overexpression of indicated plasmids.

**f** Quantification of polyubiquitinated NGN3, normalized by input NGN3. The bar graph represents mean ± SEM, and statistical significance was determined by a two-tailed Student's *t*-test (*n* = 3 biologically independent experiments).
**g** Immunoblotting for NGN3 (HA-NGN3[WT] and HA-NGN3[6SA]) and USP7 (Flag) in an HA-NGN3 immunoprecipitation experiment in HEK293A cells. **h** Quantification of immunoprecipitated Flag-USP7[WT], normalized by NGN3 output and USP7 (Flag) input from the corresponding sample. The bar graph represents mean ± SEM, and statistical significance was determined by a two-tailed Student's *t*-test (*n* = 5 biologically independent experiments). **i** Immunoblotting for exogenous NGN3 (HA-NGN3[WT] and HA-NGN3[6SA]) in TUBE affinity purification assays with or without Flag-USP7[WT] co-transfection in HEK293A cells. **j** Quantification of polyubiquitinated NGN3 (HA-NGN3[WT] and HA-NGN3[6SA]), normalized by input NGN3. The bar graph represents mean ± SEM, and statistical significance was determined by a two-tailed Student's *t*-test (*n* = 3 biologically independent experiments). Source data are provided as a Source Data file.

no gross abnormalities in *Pdx1-Cre*; *Usp7*[F/F] E14.5 pancreata (Fig. 4b). However, upon further characterization, we observed distinct populations of PDX1[low] and PDX1[high] expressing cells in the *Usp7*[F/F] E14.5 pancreas, with the PDX1[high] population largely absent in the *Pdx1-Cre*; *Usp7*[F/F] E14.5 pancreas (Fig. 4c). These distinct populations were not observed at E12.5 (Supplementary Fig. 4b, c). In line with previous studies[33], the PDX1[high] population consisted mainly of beta cells that co-stained with INS, which were significantly diminished (over a 90% reduction in INS + beta cells) in the *Pdx1-Cre*; *Usp7*[F/F] pancreas[13] (Fig. 4g). This decrease in beta-cell numbers confirms that the phenotype observed in adult mice, consisting of a reduction in the size of the endocrine compartment of the pancreas, is already present at E14.5.

We next measured expression levels of NGN3 and found that at E14.5 *Pdx1-Cre*; *Usp7*[F/F] embryos showed a dramatic reduction (over 80%) in the proportion of NGN3+ cells in the pancreas (Fig. 4c, d). RNAScope in situ hybridization carried out in E14.5 mouse embryonic pancreas showed no significant change in the % of *Neurog3* mRNA+ cells in *Pdx1-Cre*; *Usp7*[F/F] E14.5 embryos when compared to control, suggesting that the decrease in NGN3 upon *Usp7* loss was due to post-transcriptional regulation (Supplementary Fig. 5a). Surprisingly, no significant difference in NGN3+ cell numbers was detected at E12.5 (Supplementary Fig. 4b). However, as E12.5 is an early timepoint of the second wave of NGN3 protein expression (E12–E18.5), these findings may suggest that the effect of *Usp7* loss on NGN3 protein levels is compounded over-time and that USP7 is required during the later stages of pancreatic endocrine development. Moreover, staining for the proliferation marker Ki67 at E12.5 revealed no difference in the number of proliferative pancreatic cells between genotypes (Supplementary Fig. 4c), suggesting that the decrease in NGN3+ cells at E14.5 is unlikely to be caused by an impaired proliferation of pancreatic progenitors during the earlier stages of development.

Subsequently, we found a 35% reduction in cells expressing NKX2.2 in the pancreas of *Pdx1-Cre*; *Usp7*[F/F] E14.5 embryos (Fig. 4e, f), which is consistent with previous reports showing that NGN3 loss leads to reduction but not ablation of NKX2.2 expression[10]. Furthermore, we observed a drastic reduction in cells expressing INS or CHGA, a pan-endocrine marker downstream of NGN3 (Fig. 4g–i). However, we detected no significant differences in the proportion of AMY+ acinar cells or osteopontin+ (OPN+) ductal cells between the two genotypes (Supplementary Fig. 5b, c). These results recapitulate our findings in adult mice, where the endocrine, but not the exocrine, compartment is severely impacted by *Usp7* loss.

### *USP7* expression precedes *NEUROG3* expression during human pancreas development

To determine the role of *USP7* and *NEUROG3* during human pancreas development, we analyzed a publicly available single-cell RNAseq dataset generated from pancreatic tissue harvested at different timepoints throughout human fetal development[34]. After quality

control steps (Supplementary Fig. 6a–d), our dataset contained 28,073 cells. Following principal component analysis (PCA), Uniform Manifold Approximation and Projection (UMAP) was performed, which revealed distinct clusters (Supplementary Fig. 6b, c). The identity of these clusters was allocated based on the expression of bona fide markers for Acinar (*RBPJL*+), Beta (*INS*+), Trunk/progenitor (*HES1*[high], *SAT1*[low]), Tip (*RBPJL*[high], *GP2*[low]), Alpha (*GCG*+), Ductal (*SAT1*[high], *HES1*[low]), Delta (*SST*+), Gamma (*PPY*+), and Endocrine progenitors (*NEUROG3*+) (Fig. 5a and Supplementary Fig. 6e). Analysis of the UMAP with the different gestational timepoints further demonstrated the expected cell composition for each gestational point: with undifferentiated cells (trunk, tip progenitors) detected at early gestational points, and differentiated embryonic cells (beta, delta, and alpha) detected at later gestational points (Fig. 5b). Relative expression UMAP for *NEUROG3* showed an enrichment within the endocrine progenitor (EP) cluster, and scattered cells within the embryonic delta, beta and alpha clusters (Supplementary Fig. 6f). *USP7* expression was spread across the different clusters (Supplementary Fig. 6g). Co-expression UMAP revealed a number of cells expressing both *USP7* and *NEUROG3* (Fig. 5c). Interestingly, expression dot plots split by gestational weeks in the endocrine and endocrine progenitor clusters (EP, alpha, beta, delta, and pancreatic polypeptide) demonstrated that *USP7* is expressed during the *NEUROG3* expression window (Fig. 5d), peaking at W14 and preceding the *NEUROG3* expression peak (W16). To further explore the expression of *NEUROG3* and *USP7* in the endocrine compartment and determine the level of co-expression of both genes during human embryonic development, we extracted beta, alpha, delta, PP, and EP cells in a single object and generated UMAP, relative expression UMAPs and dot plots for *USP7* and *NEUROG3* (Fig. 5e–g). While USP7 was expressed across all the endocrine clusters, *NEUROG3* was restricted to EP and some scattered beta cells (Fig. 5f). Co-expression of USP7 and *NEUROG3* was observed in a subset of *NEUROG3*+ cells (Fig. 5f, g). Further analysis of the expression of *USP7*, *NEUROG3*, and *PDX1* in EP, beta, alpha, delta, and PP cell clusters represented as an expression dot plot confirmed that the EP cluster contained an overall higher average expression of *USP7* when compared to the other endocrine clusters (Fig. 5g), albeit the percentage of cells co-expressing *USP7* and *NEUROG3* was relatively small, suggesting that only a subset of *NEUROG3*+ cells co-expresses *USP7*. Interestingly, we observed that the percentage of *USP7*-expressing cells varies during the different gestational weeks in *NEUROG3*+ cells but not in *NEUROG3*- cells (Fig. 5h, i). While the % of *USP7*-expressing cells is maintained constant in *NEUROG3*− cells across different gestational weeks, the percentage of *USP7* expressing cells in *NEUROG3*+ cells increases gradually until reaching more than 30% at gestational week 14, preceding the *NEUROG3* peak and coinciding with the beta-cell generation window of human beta cells[35]. These data indicate that, similarly to its role in murine pancreas development, the USP7/NGN3 axis could be

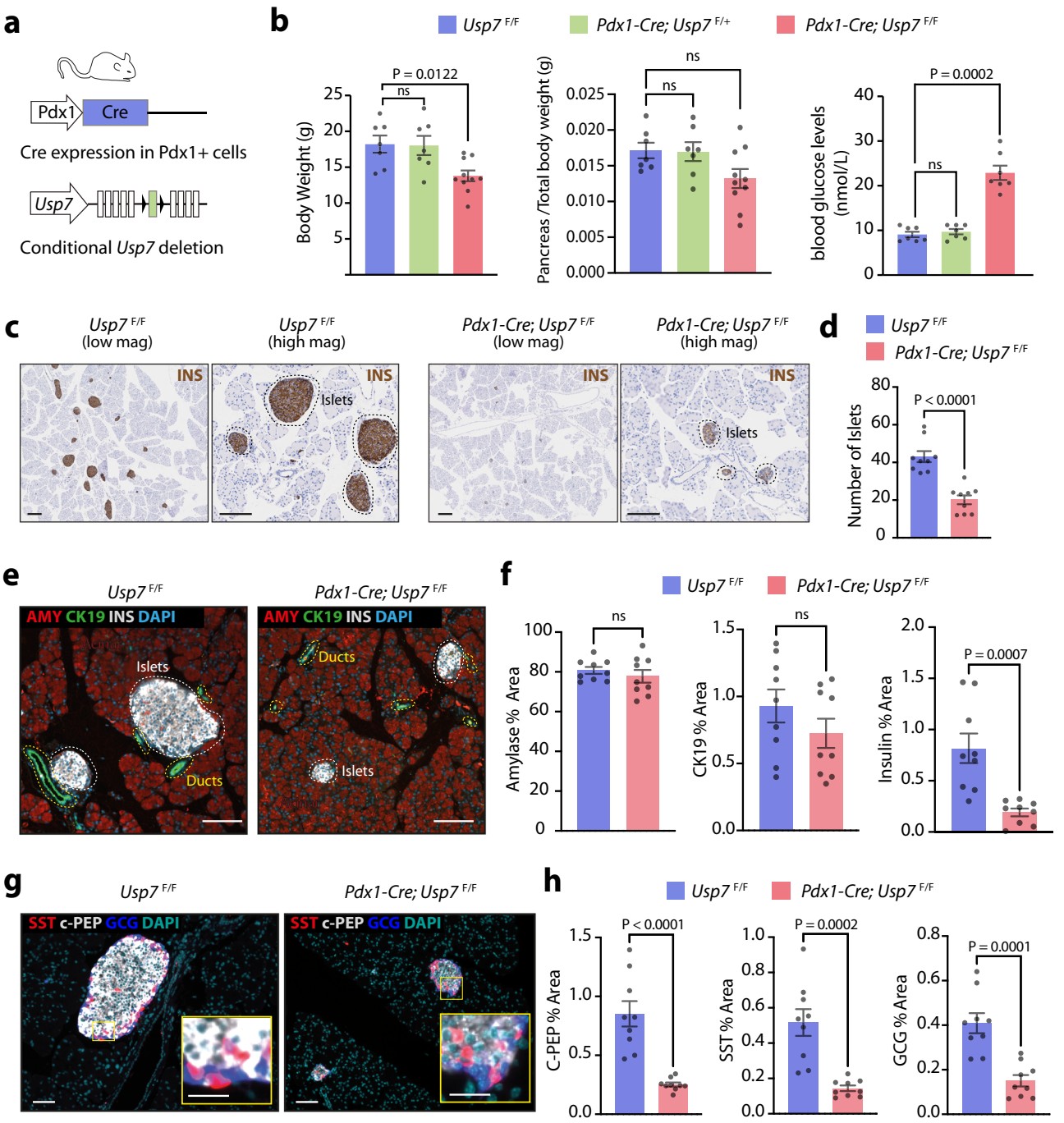

**Fig. 3 | USP7 deletion in the pancreas leads to reduced Islet formation and endocrine lineage cell differentiation. a** Schematic of pancreatic-specific *Usp7* knockout mouse. This diagram is adapted from ref. 56 Fig. 2a under Public License CC BY 4.0 (https://creativecommons.org/licenses/by/4.0/legalcode). **b** In 5–8-week-old mice, whole body weight, pancreas weight/total body weight ratio, and circulating blood glucose concentrations were measured in the indicated murine strains. The bar graph represents mean ± SEM and statistical significance was determined by one-way ANOVA with Dunnett post hoc test (*n* = 7 biologically independent animals per group, except for the *Pdx1-Cre; Usp7^F/F* group, where body weight and pancreas/total body weight was measured in ten animals). **c** Immunohistochemical analysis of Insulin (INS) expression in *Usp7* wild-type (*Usp7^F/F*) and knock-out (*Pdx1-Cre; Usp7^F/F*) pancreatic tissues. The scale bar is 200 μm in low magnification and 100 μm in high magnification. **d** Total number of islets in whole pancreatic sections in *Usp7* wild-type and knock-out mice. The bar graph represents mean ± SEM, and statistical significance was determined by a two-

tailed Student's *t*-test (*n* = 9 biologically independent animals per group). **e** Representative images of immunofluorescent staining of Amylase (AMY; red), CK19 (green), INS (white), and DAPI (blue) in *Usp7* wild-type and knock-out pancreatic tissues. The scale bar is 100 μm. **f** Quantification of positive area staining of AMY, CK19, and INS in indicated murine stains. The bar graph represents mean ± SEM and statistical significance was determined by a two-tailed Student's *t*-test (*n* = 9 biologically independent animals per group). **g** Representative images of immunofluorescent staining for c-peptide (c-PEP; white), somatostatin (SST; red), glucagon (GCG; blue), and DAPI (teal) in *Usp7* wild-type and knock-out pancreatic tissues. The scale bar is 50 μm and the insert is 100 μm. **h** Quantification of positive area staining for c-PEP, STT, and GCG in indicated murine strains. Bar graphs represent mean ± SEM and statistical significance was determined by a two-tailed Student's *t*-test (*n* = 9 biologically independent animals per group). Source data are provided as a Source Data file.

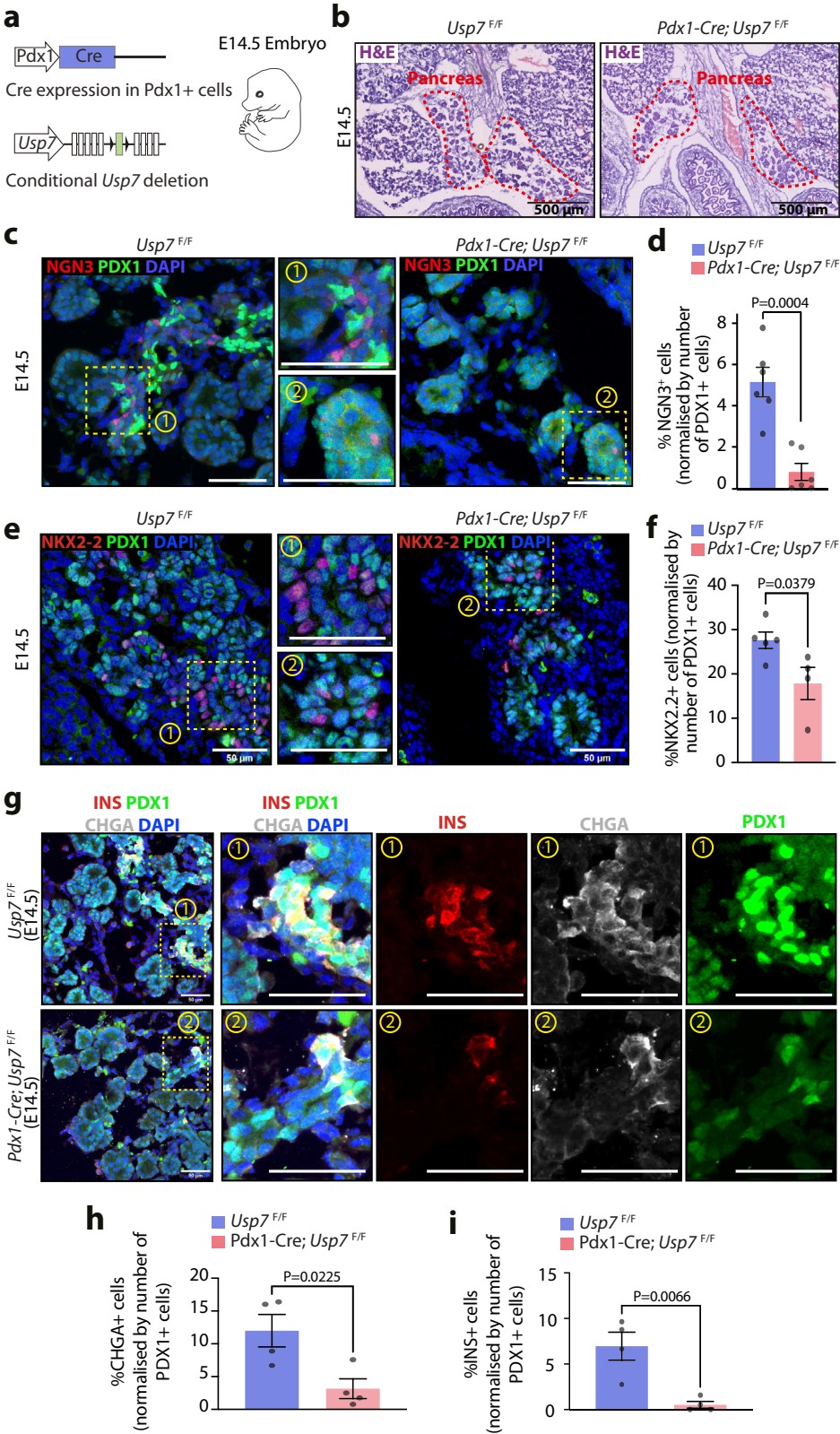

operating on a subset of *NEUROG3+* cells during the embryonic development of the human pancreas.

## USP7 inhibition impairs beta-cell differentiation in hiPSC-derived pancreas organoids

As *Usp7* deletion leads to NGN3 destabilization and a hyperglycemic phenotype in mice, we next investigated whether USP7 plays a

conserved role in human endocrinogenesis. To do this, we developed an in vitro 3D human induced-pluripotent stem cell (hiPSC) beta-cell differentiation protocol (Fig. 6a) based on a previously published protocol by refs. 36,37. To determine if USP7 plays a role in human iPSC differentiation from pancreatic progenitors to endocrine progenitors and beta cells, we treated cells with a small molecule inhibitor of USP7, termed GNE-6640[38], during the NGN3 expression window

**Fig. 4 | USP7 plays a key role in the pancreatic endocrine compartment during both mouse and human embryonic development. a** Schematic of pancreatic-specific *Usp7* knock-out time mated E14.5 embryos. This diagram is adapted from ref. 56 Fig. 2A under Public License CC BY 4.0 (https://creativecommons.org/licenses/by/4.0/legalcode). **b** Representative images of histological hematoxylin and eosin (H&E) analysis of *Usp7* wild-type (*Usp7*[F/F]) and knock-out (*Pdx1-Cre*; *Usp7*[F/F]) embryonic pancreatic tissues. The scale bar is 500 μm (*n* = 3 biologically independent embryos per genotype). **c** Immunofluorescent staining for NGN3 (red) and PDX1 (green) in the pancreas of E14.5 *Usp7*[F/F] and *Pdx1-Cre*; *Usp7*[F/F] mouse embryos. The scale bar is 50 μm. **d** Quantification of NGN3[+] cells in *Usp7*[F/F] and *Pdx1-Cre*; *Usp7*[F/F] mouse embryos, normalized by the number of PDX1[+] cells (*n* = 6 biologically independent embryos per genotype with five frames quantified for each embryo). The bar graph represents mean ± SEM, and statistical significance was determined by unpaired two-tailed Student *t*-test. **e** Immunofluorescent

staining for NKX2-2 (red) and PDX1 (green) in the pancreas of E14.5 *Usp7*[F/F] and *Pdx1-Cre*; *Usp7*[F/F] mouse embryos. The scale bar is 50 μM. **f** Quantification of NKX2-2[+] cells in *Usp7*[F/F] and *Pdx1-Cre*; *Usp7*[F/F] mouse embryos, normalized by the number of PDX1[+] cells. (*n* = 5 *Usp7*[F/F] and 4 *Pdx1-Cre*; *Usp7*[F/F] biologically independent embryos with five frames quantified for each embryo). The bar graph represents mean ± SEM, and statistical significance was determined by unpaired two-tailed Student *t*-test. **g** Immunofluorescent staining for INS (red), PDX1 (green), and CHGA (white) in the pancreas of E14.5 *Usp7*[F/F] and *Pdx1-Cre*; *Usp7*[F/F] mouse embryos. The scale bar is 50 μM. **h, i** Quantification of CHGA[+] cells or INS[+] cells in *Usp7*[F/F] and *Pdx1-Cre*; *Usp7*[F/F] mouse embryos, normalized by the number of PDX1[+] cells. (*n* = 4 biologically independent embryos per genotype with five frames quantified for each embryo). The bar graph represents mean ± SEM, and statistical significance was determined by unpaired two-tailed Student *t*-test. Source data are provided as a Source Data file.

(days 5–12). At the end of the EP stage (day 12), we detected ~15% NGN3+ cells in the untreated samples, which was dramatically reduced (more than 60%) upon GNE-6640 treatment (Fig. 6b, c).

We next assessed cell-type differentiation post the EP stage (Day 20) and found that USP7 inhibition during the NGN3 expression window led to a significant reduction (over 50%) in the number of INS+ beta-like cells. While a reduction trend in the number of GCG+ alpha-like and SST+ delta-like cells was observed with the GNE-6640 treatment, this was not statistically significant (Fig. 6d, e). Importantly, USP7 inhibition caused no significant changes in *NEUROG3* expression at the different timepoints analyzed (Fig. 6f). Concomitant with the decrease in INS+ cells at day 20, gene expression analysis showed decreased *INS* and *GCG* expression in GNE-6640-treated samples compared to control (Fig. 6g, h), despite no significant reduction in *SST* expression (Fig. 6i).

In order to determine if the GNE-6640 effect on NGN3+ and INS+ cell reduction was protocol-specific and to further validate our findings, we repeated our iPSC-to-beta-cell differentiation experiments, this time by adapting a protocol from ref. 39, with modifications by ref. 40 in our 3D iPSC-derived PP organoids (Fig. 7a). We treated the cells with GNE-6640 at different stages: Pancreas progenitors (PP) (Day 0–Day 5), endocrine progenitor (EP) stage (Day 5–Day 9), endocrine maturation stage (Day 9–Day 16), and pancreas progenitor (PP)+ endocrine progenitor (EP) stages (Day 0–Day 9) (Fig. 7a) and measured the effect on Endocrine progenitors (NGN3+) and endocrine cells (INS+ or SST+) differentiation (Fig. 7b–e). Consistent with our previous results, we observed that USP7 inhibition during the EP stage (days 5–9) led to a significant decrease in NGN3+ cells (Fig. 7b, c). Interestingly, inhibiting USP7 during the PP stage (Day 0–5) had no effect on the % of NGN3+ cells by the end of the endocrine induction stage. However, when cells were treated with GNE-6640 during both PP & EP stages (Day 0–9), the % of NGN3+ cells was significantly reduced. Accordingly, the % of INS+ cells at the end of the endocrine maturation stage was not affected when cells were treated with GNE-6640 during the PP stage (Day 0–5) or during the Endocrine maturation stage (Day 9–16), but a clear decrease in the % of INS+ cells was observed when cells were treated with GNE-6640 during the EP stage (Day 5–9) (Fig. 7d, e). These results indicate that USP7 inhibition at the PP stage, when NGN3 is still not present, does not impact the generation of NGN3+ endocrine progenitors or INS+ cells. Conversely, inhibiting USP7 in stages where NGN3 is present (EP) results in a clear reduction of endocrine progenitors and INS+ cells. Moreover, there was no significant change observed in the % of SST cells upon GNE-6640 treatment, consistent with our previous differentiation protocol (Fig. 6e). All together, these data suggest that USP7 activity during the EP stage is essential to the generation of human iPSC-derived endocrine progenitors as well as beta-like cells and that USP7 inhibition in a human in vitro model for beta-cell differentiation results in impaired NGN3 stability and reduced beta-cell generation.

## Discussion

Neurogenin 3 is essential and sufficient for endocrine specification during pancreas development. Achieving high levels of NGN3 expression at specific times during embryogenesis is a requirement for the generation of all pancreatic endocrine cell types, including beta cells[17,41]. In this study, we identified USP7 as an interactor of NGN3 and a key regulator of endocrine cell fate choice in the developing pancreas. We found that USP7 interacts with and deubiquitinates NGN3 in vitro, leading to its stabilization. Other studies have previously identified transcriptional regulators of NGN3, such as SOX9[18], HNF6[19], HNF1α[20], FOXA2[20], and PDX1[21]. Additionally, the NOTCH/HES1 pathway plays a role in NGN3 post-translational regulation, destabilizing NGN3 and limiting endocrine differentiation[29]. While we have previously shown that NGN3 is targeted to the proteasome for degradation by the ubiquitin ligase FBW7[24], we have now expanded on this mechanism by identifying USP7 as the first deubiquitinase known to interact with NGN3 and play a key role in its regulation during pancreatic development and homeostasis. In contrast to its interaction with FBW7, the interaction between NGN3 and USP7 does not seem to be dependent on phosphorylation at individual predicted phosphorylation sites within NGN3 that contain a serine-proline motif, but rather by the overall phosphorylation on NGN3. This is likely due to low phosphorylation levels leading, in turn, to low NGN3 ubiquitination, which then translates to a reduction in NGN3/USP7 interactions.

Other established substrates of USP7, such as p53, PHF8, and PLK1, play important roles in regulating cell cycle and proliferation[27,42–44]. As deletion of *Usp7* in the mouse embryonic pancreas leads to a reduction in islet number and size with no apparent effect on the ductal or acinar compartments, the observed phenotype is unlikely to result from defects in the proliferation or expression of PDX1 in progenitors that give rise to both the endocrine and exocrine compartment. Indeed, staining for the proliferation marker KI67 in E12.5 murine sections revealed no significant difference in the number of proliferative cells in the *Pdx1-Cre*; *Usp7*[F/F] embryonic pancreas when compared to controls. Furthermore, while *Usp7* loss led to a significant decrease in NGN3+ endocrine progenitors at E14.5, the % of cells expressing *Neurog3* mRNA in the pancreas of *Pdx1-Cre*; *Usp7*[F/F] embryos was not significantly decreased compared to controls at E14.5. This suggests that the observed decrease in NGN3+ cells in the embryonic pancreas upon *Usp7* deletion is due to a post-translational mechanism.

Our results show that the reduction of NGN3+ progenitors at the embryonic stage results in the impairment of endocrine differentiation in the adult *Pdx1-Cre*; *Usp7*[F/F] mice. However, consistent with the existence of a few residual NGN3+ cells at E14.5 in *Pdx1-Cre*; *Usp7*[F/F] embryos, we do observe some endocrine cells at the adult stages. One plausible explanation is that compensatory mechanisms (other DUBs) could counteract the loss of USP7. Alternatively, the scattered NGN3+ cells found at E14.5 in *Pdx1-Cre*; *Usp7*[F/]embryos could be escapers from the Cre-mediated recombination. Nevertheless, the residual NGN3+ and INS+ cells in the *Pdx1-Cre*; *Usp7*[F/F] pancreas were not able to

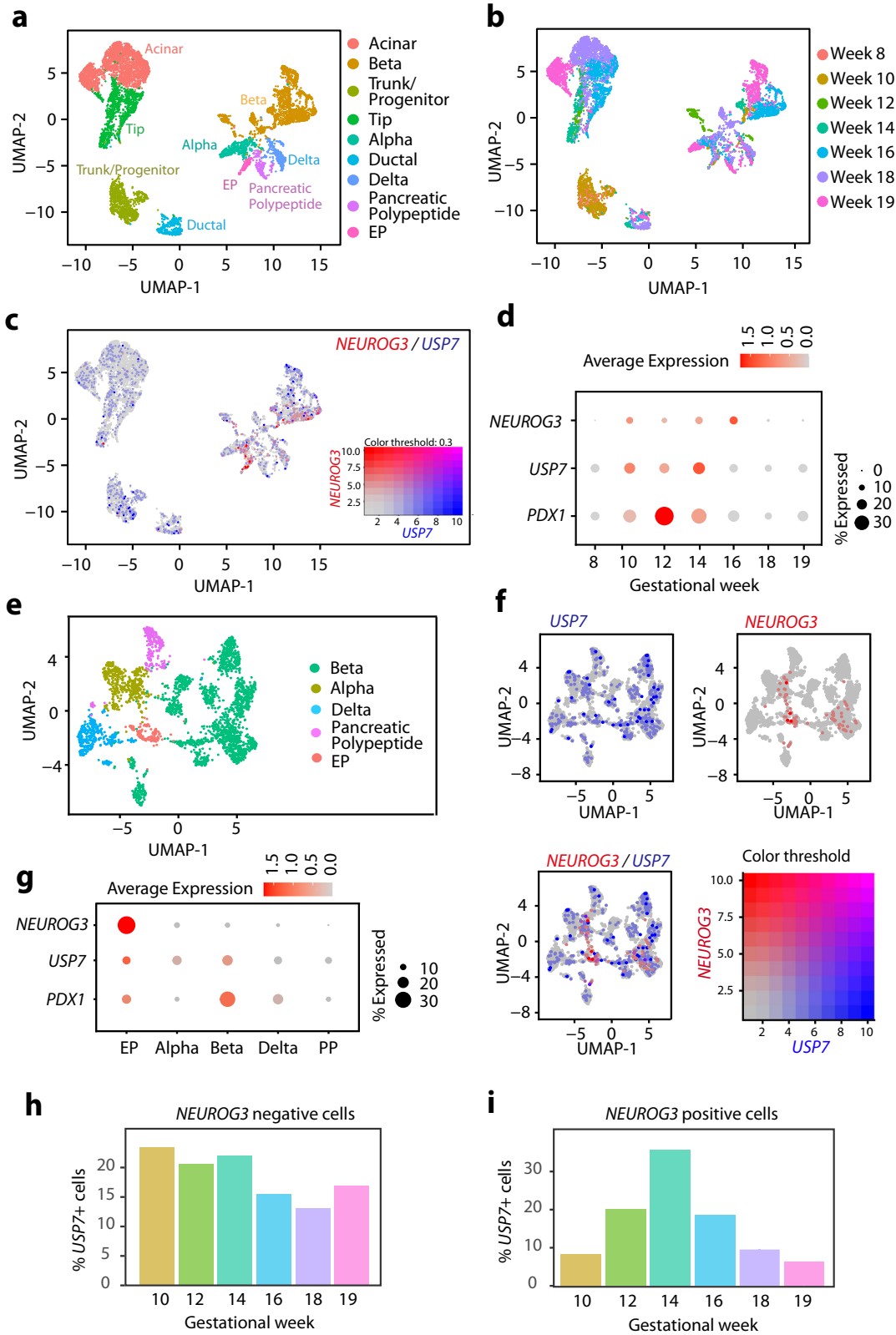

**Fig. 5 | *USP7* expression during human embryonic pancreas development.**
**a** UMAP projection of pancreatic epithelial cells in OMIX236 human fetal pancreas dataset. Cells are colored by assigned clusters. **b** UMAP projection of pancreatic epithelial cells. Cells are colored by developmental timepoints. **c** Co-expression of *NEUROG3* (red) and *USP7* (blue) across the scRNA-seq dataset. **d** Expression of *NEUROG3*, *USP7*, and *PDX1* across developmental timepoints in *NEUROG3*-expressing clusters (EP, alpha, beta, delta, and pancreatic polypeptide) in the OMIX236[34] scRNA-seq dataset. **e** UMAP projection of endocrine cells in OMIX236 human fetal pancreas dataset. Cells are colored by assigned clusters. **f** Co-expression of *NEUROG3* (red) and *USP7* (blue) across the endocrine clusters of the scRNA-seq dataset. **g** Expression of *NEUROG3, USP7*, and *PDX1* across each *NEUROG3*-expressing cluster in the endocrine clusters of the scRNA-seq dataset. **h** Percentage of USP7+ cells across developmental timepoints in all *NEUROG3*-negative cells of the endocrine clusters. **i** Percentage of *USP7*+ cells across developmental timepoints in all *NEUROG3*-positive cells of the endocrine clusters.

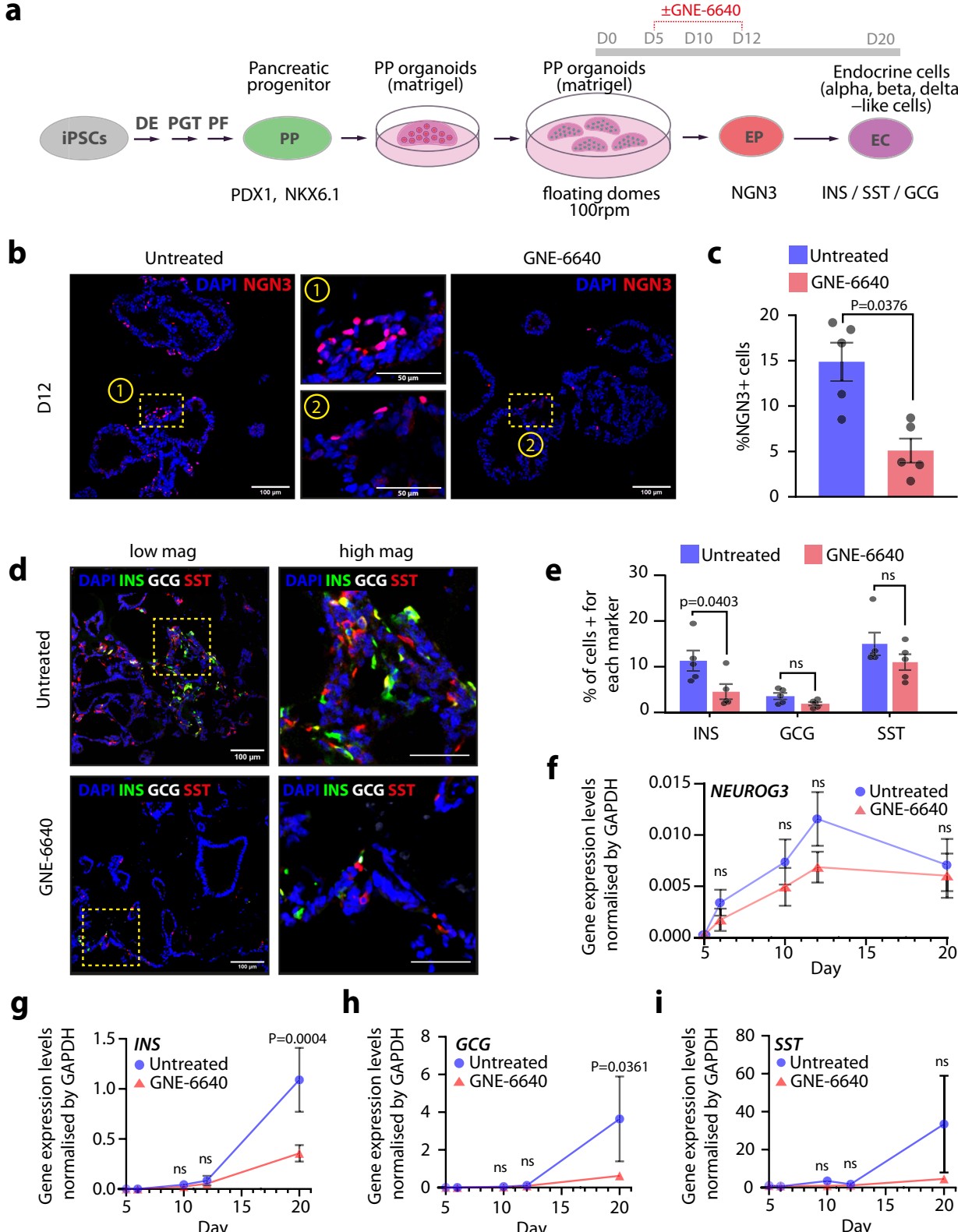

prevent weight decrease (growth retardation) and hyperglycemia, which are consistent with an INS knock-out phenotype[45].

The fact that depletion of *Usp7* leads to a reduction in NKX2.2+, CHGA+, and INS+ cells, while not affecting the acinar and ductal compartments, is consistent with the previously reported phenotype of an *Neurog3* deletion model[12]. While these *Usp7* loss-mediated effects are already visible at E14.5, leading to a reduction in NGN3+ endocrine

progenitor numbers, we could not exclude an additional effect of *Usp7* depletion on endocrine cell differentiation after this stage. However, our in vitro experiments in which we inhibit USP7 activity, specifically in the endocrine progenitor stage, support our in vivo findings that loss of USP7 diminishes both NGN3+ endocrine progenitors and INS + beta-like cells. We acknowledge that the efficiency of differentiation is slightly reduced in our 3D differentiation protocol compared to cells

**Fig. 6 | USP7 inhibition impairs endocrine specification during iPSC-derived PP organoids to-beta-cell differentiation. a** Schematic of the adapted differentiation protocol, with or without GNE-6640 treatment during the endocrine specification stage (D5–D12). This diagram is adapted from ref. 57 Fig. 1a, d under Public License CC BY 4.0 (https://creativecommons.org/licenses/by/4.0/legalcode). **b** NGN3 (red) immunofluorescent staining in control and GNE-6640-treated samples at D12. The scale bar is 100 μm (low magnification) and 50 μm (high magnification). **c** Quantification of NGN3⁺ cells in D12 control and GNE-6640-treated samples. The bar graph represents mean ± SEM, and statistical significance was determined by unpaired two-tailed Student *t*-test (*n* = 5 biologically independent experiments). **d** Immunofluorescent staining for INS, GCG, and SST in D20 control and GNE-6640-treated samples. The scale bar is 100 μm (low magnification) and 50 μm (high magnification). **e** Quantification of INS+, GCG+, and SST+ cells in control and GNE-

6640-treated samples. The bar graph represents mean ± SEM, and statistical significance was determined by unpaired two-tailed Student *t*-test (*n* = 5 biologically independent experiments). **f** *NEUROG3* gene expression normalized by *GAPDH* in GNE-6640-treated and control samples. The plot represents mean ± SEM, and statistical significance was determined by two-way ANOVA with Sidak multiple comparison correction; Comparison against control sample from the same timepoint (*n* = 4 biologically independent experiments). **g**–**i** *INS*, *GCG*, and *SST* gene expression normalized by *GAPDH* in GNE-6640-treated and control samples (*n* = 3 biologically independent experiments). Bar graphs represent mean ± SEM, and statistical significance was determined by two-way ANOVA with Sidak multiple comparison correction; Comparison against control sample from the same timepoint. Source data are provided as a Source Data file.

in suspension[46,47]. Yet, our protocols allow for comparison between conditions in a setting where the cells are receiving 3D physical cues more similar to the in vivo state than when cells are grown in suspension, and where the % of NGN3+ cells is more similar to the one observed in vivo. It is important to note that we did not observe significant changes in either GCG+ or SST+ cell populations in our human iPSC-based models, yet both populations were significantly reduced in vivo upon *Usp7* deletion. This may reflect the disproportionate effects of this inhibition on distinct endocrine progenitor pools[17,48], leading to the beta-like population being more severely impacted than alpha- or delta-like cells. To address this, we carried out further differentiation experiments, inhibiting USP7 at different stages of pancreatic-progenitor-to-beta-cell differentiation. However, regardless of the timing, or duration of USP7 inhibition, we did not observe significant reductions in SST+ delta-like cells. While this could be due to intrinsic differences in pancreatic development and NGN3 regulation between mouse and human systems, we cannot exclude the possibility that our in vitro system does not fully recapitulate in vivo GCG and SST development. This may account for the more drastic effect of *Usp7* knockout in vivo, compared to our human iPSC model. Despite this limitation, impairing USP7 function consistently results in decreased NGN3+ endocrine progenitor numbers, followed by a reduction in beta-cell generation in both mouse and human systems, suggesting an essential role for USP7 in pancreatic endocrine development.

Together, our results identified the NGN3-USP7 axis as a fundamental mechanism that controls NGN3 stability to ensure the correct specification of endocrine cells in the developing pancreas. Despite this finding, how USP7 itself is regulated in the context of pancreas development, both transcriptionally and post-translationally, has not been fully described. Additional research into the regulation of USP7 could help reveal new pathways modulating the NGN3-USP7 axis during the NGN3 expression window, offering insight into the mechanisms that push pancreatic progenitors towards an endocrine fate, with potential uses for in vitro and in vivo beta-cell generation for diabetes therapy.

## Methods
### Mice
All animal experiments were approved by the Francis Crick Institute and Institute of Cancer Research Animal Ethics Committees and conformed to UK Home Office regulations under the Animals (Scientific Procedures) Act 1986, including Amendment Regulations 2012. The *Usp7*[F/F][31] and *Pdx1-Cre*[49] mouse lines have been previously described. These lines were inter-crossed on a C57BL/6 background to generate the genotypes of this study. All strains were genotyped by Transnetyx. The mice were housed in a constant temperature, humidity, and pathogen-free controlled environment (23 ± 2 °C, 50–60%) cage with a standard 12 h light/12 h dark cycle, plenty of water, and food (pathogen-free) in their cage. All animal experiments were conducted in adults, aged 5–8 weeks. Where indicated, blood glucose measurements were taken at the endpoint on whole blood using FreeStyle

Optium Neo (Abbott) glucose meter following the manufacturer's instructions.

### iPSC differentiation
Human iPSCs from the Kute-4 line were plated into a 12-well plate at 500,000 cells/well seeding density and were differentiated into pancreatic progenitors for a period of 14 days using the STEMdiff™ Pancreatic Progenitor Kit (Stem Cell Technologies, 05120) following manufacturer's instructions. Cells were then encapsulated into Matrigel (Corning, 356230), plated in 25 μl domes, and expanded following a previously established protocol[50]. Confluent domes were detached and transferred intact to a six-well plate for suspension culture on a shaker at 100 rpm and the differentiation proceeded with Stages 4–6 of the Hogrebe et al. protocol[36] or with the pancreatic progenitor-to-beta-cell stages of the modified Russ et al. protocol[39,40], with daily media changes. USP7 inhibition was carried out by supplementing the differentiation media with 5 μM GNE-6640 (AOB37854-1, Generon).

### Immunohistochemistry of adult mouse pancreas sections
Tissues were collected, fixed in 10% neutral-buffered formalin (NBF, Sigma-Aldrich) for 16 h, dehydrated in 70% ethanol, and embedded in 4 μm paraffin sections. The slides were de-paraffinized in xylene and rehydrated using a series of graded industrial methylated spirits solutions to distilled water. No antigen retrieval was performed for either USP7 (Bethyl, A300-033A) or Insulin (Merck, K36AC10) staining. Endogenous peroxidase blocking was performed using 1.6% $H_2O_2$ for 10 min at room temperature (RT) and protein blocking was performed using 2.5% Normal Horse Serum (ready-to-use; MP-7401, Vector Laboratories) overnight at 4 °C. Primary USP7 antibody was diluted 1:500, and Insulin antibody was diluted 1:2000 in 1% BSA and incubated overnight at 4 °C. After washing in PBS, slides were incubated in HRP Horse Anti Rabbit IgG Polymer (MP-7401, Vector Laboratories) or HRP Horse Anti Mouse IgG Polymer (MP-7402, Vector Laboratories) for 30 min at RT. Finally, the slides were developed in 3,3-diaminobenzidine (DAB) chromogen (SK-4100, Vector Laboratories) for 10 min at RT. The slides were counterstained with Harris Hematoxylin (3801561E, Leica Biosystems), dehydrated, cleared, and mounted in a Sakura Tissue-Tek Prisma® autostainer. Tissue slides were processed using the Zeiss Axio Scan.Z1 slide scanner with Zen 3.0 software package. Immunohistochemistry intensity quantifications were measured using ImageJ software (version 1.53k) on whole tissue sections, three whole tissue sections at different levels per biologically independent animal. Percent area staining (% Area) was calculated by conversion of image to RGB, tracing the whole tissue area, and applying the Zen threshold prior to measurement of staining intensity.

### Immunofluorescence staining of adult mouse pancreas sections
Tissues were collected and fixed, embedded, and sectioned as described above. Slides were dewaxed with Histo-Clear (National Diagnostic) and dehydrated using a series of graded industrial ethanol solutions to distilled water. Heat-mediated antigen retrieval was done

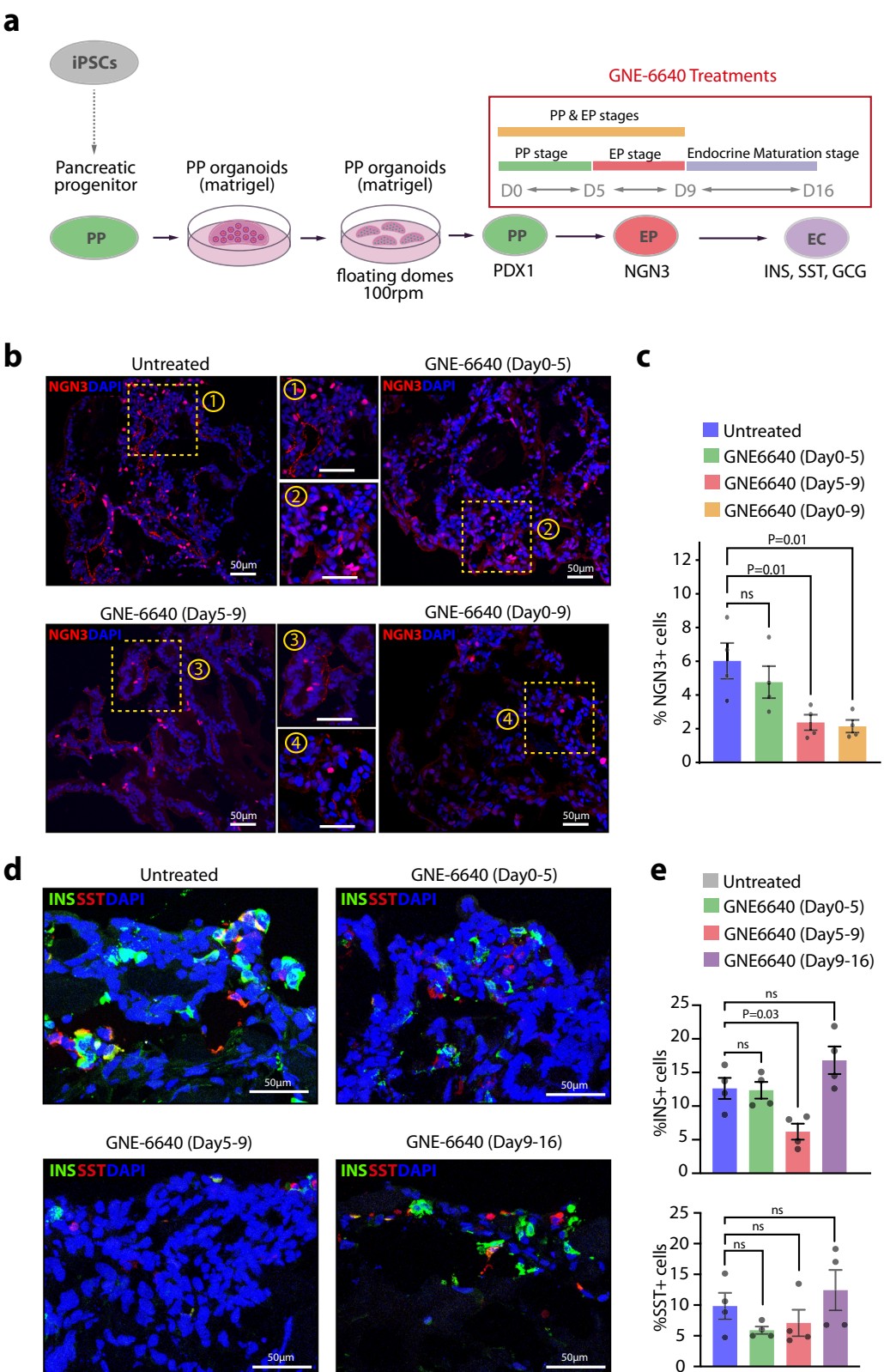

in 10 mM sodium citrate buffer (pH 6.2) and blocking of endogenous peroxidase was achieved with 1% BSA in PBS + 5% FCS. Tissues were incubated overnight with primary antibodies (Supplementary Table 1) and 0.3 μM DAPI (Sigma-Aldrich) at the described dilutions below. Following, slides were washed 3x with PBS and incubated with secondary antibodies (Supplementary Table 1) for 1 h at room temperature. Background fluorescence was blocked with Sudan black

incubation, and slides were mounted with a fluorescent mounting medium (Dako). Tissue slides were processed using the Zeiss Axio Scan.Z1 slide scanner with Zen 3.0 software package. Fluorescence intensity quantifications were measured using ImageJ software (version 1.53k) on whole tissue sections, three whole tissue sections at different levels per biologically independent animal. Percent area staining (% Area) was calculated by conversion of image to RGB,

**Fig. 7 | USP7 inhibition during the pancreatic progenitor and endocrine maturation stages of iPSC-to-beta-cell differentiation does not impact INS+ beta-like cell generation. a** Schematic of the adapted differentiation protocol, with or without GNE-6640 treatment during the pancreatic progenitor (D0–D5), endocrine progenitor (D5–D12), or maturation stage (D12–D16). This diagram is adapted from ref. 57 Fig. 1a, d under Public License CC BY 4.0 (https://creativecommons.org/licenses/by/4.0/legalcode). **b** NGN3 (red) immunofluorescent staining in control and GNE-6640-treated samples at D12. The scale bar is 50 µm. **c** Quantification of NGN3+ cells in D12 control and GNE-6640-treated samples. The bar graph represents mean ± SEM, and statistical significance was determined by one-way ANOVA with the Dunnett post hoc test (n = 4 biologically independent experiments). **d** Immunofluorescent staining for INS, GCG, and SST in D16 control and GNE-6640-treated samples. The scale bar is 50 µm. **e** Quantification of INS+, GCG+, and SST+ cells in control and GNE-6640-treated samples. The bar graph represents mean ± SEM, and statistical significance was determined by one-way ANOVA with the Dunnett post hoc test (n = 4 biologically independent experiments). Source data are provided as a Source Data file.

tracing the whole tissue area, and applying the Zen threshold prior to measurement of staining intensity.

## Immunostaining of mouse embryo sections

The trunks of E14.5 and E12.5 wildtype and *Pdx1-Cre; Usp7*[F/F] mouse embryos were fixed in 4% paraformaldehyde (PFA) at 4 °C, O/N, washed in PBS, equilibrated in 20% sucrose for 24 h at 4 °C, and embedded into OCT (361603E, Thermo Fisher Scientific). The blocks were then sectioned longitudinally, and frozen sections (10 µm thickness) containing the embryonic pancreas were collected. Section slides underwent heat-mediated antigen in 10 mM sodium citrate buffer (pH 6.2), followed by permeabilization in 0.03% Triton X-100 for 5 min and blocking for 1 h at RT in PBS with 10% fetal bovine serum (FBS), 3% bovine serum albumin (BSA), 0,05% Triton X-100, 0,05% Tween 20, and 0.25% fish gelatin. The sections were then stained with primary antibodies in the dark, at 4 °C, O/N, washed in PBS, then stained with secondary antibodies and DAPI (2.5 µg/ml) in the dark for 1 h at RT. Slides were washed three times with PBS, mounted with ProLong™ Gold Antifade Mountant (Thermo Fisher Scientific, P36934), and imaged on a Leica SP8 confocal microscope at 40x magnification, with LAS X software. For each embryo, at least 3 representative images of pancreatic tissue (delineated by PDX1 expression) were used for quantification. Counting of NGN3+, NKX2.2+, CHGA+, INS+, OPN+, AMY+, KI67+, and PDX1+ cells was carried out using ImageJ software (Cell Counter function). The proportion of cells positive for each marker was assessed by calculating the percentage of these cells out of all pancreatic (PDX1+) cells in the frame.

## RNAScope in situ hybridization of mouse embryo sections

E14.5 wildtype and *Pdx1-Cre; Usp7*[F/F] mouse embryo cryosections were prepared as previously described. RNA in situ hybridization was performed on cryosections using the RNAscope Multiplex Fluorescent Reagent Kit v2 (Advanced Cell Diagnostics, 323136) according to the manufacturer's protocol. The sections were permeabilized with Protease III for 20 min at 40 °C in the HybEZ oven (Advanced Cell Diagnostics, 321710/321720). The Mm-*Neurog3*-C1 (Advanced Cell Diagnostics, 422401) probes were used to detect *Neurog3* mRNA, and sections were developed with Opal 570 fluorophores (Akoya, FP1488001KT). Images were acquired on a Leica SP8 Confocal microscope using 63x or 40x objective lens, using LAS X software. For each embryo, three representative images were used for quantification. Counting of *Neurog3*+ cells was carried out using ImageJ software (Cell Counter function). The percentage of *Neurog3*+ cells was calculated out of all pancreatic cells in the frame based on visible pancreatic morphology.

## Cell culture and reagents

HEK293A cells were grown in DMEM (Gibco, 61965026) supplemented with 10% fetal bovine serum (FBS) and 1% penicillin/streptomycin. Cells were passaged weekly with TryplE (Gibco, 12604021). iPSC-derived pancreatic progenitor organoids were generated as described before[37] and cultured in Matrigel (Corning, 356230) following previously established protocols[50,51]. Cells were passaged every 7–10 days by dissolving the Matrigel in TryplE (5 min, 37 °C, on a shaker). After organoid dissociation, organoids were resuspended in Matrigel and grown in a medium containing Advanced DMEM/F-12, 10 mM HEPES, 1X Glutamax supplement, 1% Penicillin/Streptomycin, 0.5 µM A83-01, 0.05 µg/ml mouse EGF, 0.1 µg/ml human FGF-10, 10 nM Gastrin, 1.25 mM *N*-acetylcysteine, 10 mM Nicotinamide, 1X B27 supplement, and 10% Noggin and R-spondin-conditioned-media with 10.5 µM Y-27631 (only for the first 48 h after splitting) as published by others[50,51].

## Plasmid generation and transfection procedure

The HA-NGN3[WT], HA-NGN3[S183A], HA-NGN3[S187A], HA-NGN3[6SA], Ub-His, and Flag-USP7[WT] constructs have been previously described in refs. 24,25,52. The HA-NGN3[6SA] plasmid has been kindly provided by Anna Philpott (Cambridge Stem Cell Institute). All other NGN3 mutant constructs (S14A, S38A, S160A, S174A, and S199A) were generated through conventional PCR site-directed mutagenesis from the original HA-NGN3[WT] construct. The Flag-tagged expression plasmids encoding human *USP7* were kindly provided by Vivian Li (The Francis Crick Institute). The Flag-USP7[C223A] was generated through conventional PCR site-directed mutagenesis. Transfection of HEK293A followed the previously described Polyethylenimine (PEI) transfection protocol (Longo et al. 2013), adjusted for different plate formats and plasmid DNA concentrations. Briefly, cells were plated onto tissue culture dishes just below confluency and left to attach overnight. For transfection into a 10 cm dish, 10 µg of DNA were mixed with 250 µl of Optimem (Thermo Fisher Scientific, 31985062) and incubated for 10 min. About 30 µl of PEI were simultaneously incubated in another 250 µl of Optimem, then added to the DNA mix and further incubated at room temperature for 20 min, before being pipetted dropwise onto the tissue culture dish containing the previously plated cells.

## Immunoblotting

HEK293A cells were lysed in NP-40 lysis buffer (containing 20 mM Tris pH 7.5, 150 mM NaCl, 1 mM EDTA, 0.2% IGEPAL, and 10% glycerol) supplemented with 0.1 M sodium fluoride (NEB, P0759), 10 mM Phenylmethylsulfonyl fluoride (Sigma-Aldrich, PMSF-RO), 10 mM Orthovanadate (NEB, P0758), and Protease inhibitor cocktail (Sigma-Aldrich, P8340-5ML). Protein lysates (30 µg per lane) were resolved on 10% SDS-PAGE gels and transferred to PVDF membranes. Membranes were blocked with 5% milk in 1x TBS-T buffer (Severn Biotech, 20-7310-10), blotted with primary then secondary antibodies (Supplementary Table 1), and developed using a Clarity Western ECL Substrate kit (BioRad, #1705061). Uncropped images of representative blots are available in the Source Data file.

## Protein co-immunoprecipitations

HEK293A cells were transfected with 5 µg HA-NGN3 (wildtype or mutant) plasmid DNA and 5 µg Flag-USP7 or Flag-USP7[C223A]. An empty pcDNA3 vector was used to make up the total amount of DNA added to each sample to 10 µg in HA-NGN3-only and Flag-USP7-only controls. After 18 h, cells were treated with 1 µM MG132 (Alfa Aesar, J63250.MA) for 6 h, then harvested and lysed in NP-40 buffer (as described in the "Immunoblotting" section). Protein lysates were incubated with Pierce™ Anti-HA Magnetic Beads (Thermo Fisher Scientific, 88837) on a vertical shaker, O/N, at 4 °C. The beads were then washed with NP-40 lysis buffer five times, resuspended in 1x Laemli Buffer in NP-40 buffer,

and boiled at 95 °C for 10 min. The supernatant was then separated from the beads and processed according to the previously described immunoblotting protocol.

## Ubiquitinated NGN3 TUBE affinity purification assays

HEK293A cells were transfected with 2.5 µg HA-NGN3 plasmid DNA, 3.5 µg Ub-His (wildtype, K48R or K63R) plasmid DNA, and 2.5 µg Flag-USP7 or Flag-USP7$^{C223A}$. In the case of USP7 titration experiments, varying amounts of Flag-USP7 plasmid DNA were used for transfection, from 0.5 to 2.5 µg. An empty pcDNA3 vector was used to make up the total amount of DNA added to each sample to 10 µg. After 42 h, cells were treated with 1 µM MG132 for 6 h, then harvested, lysed in NP-40 buffer (as described in "Immunoblotting" section), and the amount of protein quantified. Protein lysates were incubated with anti-ubiquitin TUBE 2 beads (LifeSensors, UM402) for 2.5 h at 4 °C on a vertical shaker. The beads were then washed with NP-40 lysis buffer five times, resuspended in 1x Laemli Buffer in NP-40 buffer, and boiled at 95 °C for 10 min. The supernatant was then separated from the beads and processed according to the previously described immunoblotting protocol.

## RT-qPCR

Pancreatic organoid domes were collected at different timepoints throughout the differentiation for both the control and the GNE-6640-treated samples. RNA was extracted using the RNeasy mini kit (Qiagen, 74106) and cDNA was prepared using QuantiTect Reverse transcription kit (Qiagen, 205314) according to the manufacturer's instructions. cDNA was then diluted with nuclease-free water to a concentration of 7.5 ng/µl. To set up a qPCR reaction, 15 ng of the sample were mixed with 5 µl of SYBR™ Green PCR Master Mix (Thermo Fisher Scientific, 4309155), 1.25 µl of a 1:1 mix of forward reverse primer (2.5 µM each) (Supplementary Table 2), and 1.75 µl of nuclease-free water. Each reaction was plated in triplicate. Plates were spun at 2000 × g for 5 min before running on a CFX 384 Touch RT-qPCR machine. Gene expression was determined by normalization to the housekeeping gene *GAPDH*.

## Immunostaining of 3D pancreatic organoids

Pancreatic organoid domes were collected throughout the differentiation and fixed in 4% PFA for 20 min at RT, washed three times with PBS, and equilibrated in 30% sucrose O/N at 4 °C. Domes were then embedded into OCT (361603E, Thermo Fisher Scientific) and sectioned into 10-µm-thick frozen sections, which were then permeabilized, blocked, stained, mounted, and imaged as previously described for mouse embryo frozen sections, without antigen retrieval. For each condition, five different frames were analyzed, with cell counting carried out through ImageJ software (Cell Counter function). The proportion of NGN3+, INS+, GCG+, and SST+ cells was calculated as a percentage of all cells in the frame, as delineated by DAPI-stained nuclei. Due to the minimal number of GCG+ cells generated at the end of the Russ et al., quantification and statistical analysis could not be carried out for the effect of USP7 inhibition on GCG+ cell numbers.

## Human embryonic pancreas scRNA-seq dataset analysis

The human foetal pancreas dataset was initially published by ref. 34 and accessed from OMIX (https://bigd.big.ac.cn/omix/; identifier OMIX236). Dataset preprocessing and analysis was performed in R 4.1.1 using the Seurat package (version 4.0.4)[53]. Cells with a fraction of mitochondrial gene counts >15% were filtered out to remove stressed and dying cells. Doublets and empty droplets were also removed by selecting cells with >500 or <4000 genes detected. A total of 28,073 cells were used for downstream analysis. Data were then normalized and scaled using the SCTransform() function with a Gamma-Poisson Generalized Linear Model. The top 3000 highly variable genes were computed and dimensionality reduction was performed via principal component analysis (PCA) with $n = 30$ dimensions using the RunPCA() function and via Uniform Manifold Approximation and Projection (UMAP) using the RunUMAP() function. A nearest-neighbor graph was constructed using the FindNeighbors() function with default settings (k value = 20). Lastly, clusters of cells were identified using the Louvain algorithm (FindClusters() function) with the resolution parameter set to 0.2. UMAP plots were used for data representation. For cell-type annotation, a Wilcoxon rank-sum test for differential gene expression was performed with the FindMarkers() function to identify the globally-enriched genes within each cluster. Each cluster was annotated based on the expression of marker genes: Epithelial (*EPCAM*$^+$), mesenchymal (*COL3A1*$^+$), endothelial (*PECAM1*$^+$), neurons (*ASCL1*+), and immune (*PTPRC*$^+$). To further resolve the cellular composition of the epithelial cluster, epithelial cells were subsetted and re-clustered as previously detailed. The newly generated clusters were then annotated based on the expression of marker genes: Acinar (*RBPJL*$^+$), Beta (*INS*$^+$), Trunk/progenitor (*HES1*$^{High}$, *SAT1*$^{low}$), Tip (*RBPJL*$^{hi}$, *GP2*$^{low}$), Alpha (*GCG*$^+$), Ductal (*SAT1*$^{high}$, *HES1*$^{low}$), Delta (*SST*$^+$), Gamma (*PPY*$^+$), and Endocrine progenitors (*NEUROG3*$^+$). To focus on the endocrine population, the Alpha, Beta, Gamma, Delta, and Endocrine progenitor clusters were subsetted, and dimensionality reduction was performed as previously described. Lastly, *NEUROG3*+ and *NEUROG3*− cells were selected based on the number of NEUROG3 counts being >0 (*NEUROG3*+) or =0 (*NEUROG3*−).

## Identification of NGN3 interactors by IP-MS

Duplicate samples transfected with pcDNA3-HA-NGN3 and a pcDNA3 empty vector control were prepared and incubated with anti-HA beads as previously described in the Protein co-immunoprecipitations section. After boiling, samples were run on a 10% Mini-PROTEAN® TGX Precast Protein Gel (BioRad, 4561033) at 100 V for 20 min, then excised out of the gel. LC-MS/MS analysis of the gel sections, as well as analysis of the raw data obtained, was carried out by the Centre of Excellence for Mass Spectrometry, King's College London.

Gel sections were prepared for in-gel reduction, alkylation, and enzymatic digestion. Cysteine residues were reduced with dithiothreitol and derivatized by treatment with iodoacetamide to form stable carbamidomethyl derivatives. Trypsin digestion was carried out overnight at room temperature after initial incubation at 37 °C for 2 h. The extracted peptide samples were individually resuspended in 18 µl of resuspension buffer (2% can in 0.05% FA), 6 µl of which was injected to be analyzed by LC-MS/MS. Chromatographic separation was performed using a U3000 UHPLC NanoLC system (Thermo-FisherScientific, UK). Peptides were resolved by reversed-phase chromatography on a 75-um C18 Pepmap column (50 cm length) using a three-step linear gradient of 80% acetonitrile in 0.1% formic acid. The gradient was delivered to elute the peptides at a flow rate of 250 nl/min over 60 min starting at 5% B (0–5 min) and increasing solvent to 40% B (5–40 min) prior to a wash step at 99% B (40–45 min) followed by an equilibration step at 5% B (45–60 min). The eluate ionized by electrospray ionization using an Orbitrap Fusion Lumos (Thermo Fisher Scientific, UK) operating under Xcalibur v4.3. The instrument was first programmed to acquire using an Orbitrap-Ion Trap method by defining a 3 s cycle time between a full MS scan and MS/MS fragmentation by collision-induced dissociation. Orbitrap spectra (FTMS1) were collected at a resolution of 120,000 over a scan range of m/z 375–1800 with an automatic gain control (AGC) setting of 4.0e5 (100%) with a maximum injection time of 35 ms. Monoisotopic precursor ions were filtered using charge state (+2 to +7) with an intensity threshold set between 5.0e3 to 1.0e20 and a dynamic exclusion window of 35 s ± 10 ppm. MS2 precursor ions were isolated in the quadrupole set to a mass-width filter of 1.6 m/z. Ion trap fragmentation spectra (ITMS2) were collected with an AGC target setting of 1.0e4 (100%) with a maximum injection time of 35 ms with CID collision energy set at 35%.

Raw mass spectrometry data were processed into peak list files using Proteome Discoverer (ThermoScientific; v2.5) (Fig. 1). The raw data file was processed and searched using the Mascot search algorithm (v2.6.0; www.matrixscience.com) and the Sequest search algorithm (Eng et al.; https://doi.org/10.1016/1044-0305(94)80016-2) against the Uniprot Human Taxonomy database (27,277 entries) at 1% FDR stringency including a decoy database search. Scaffold (version Scaffold_5.1.0, Proteome Software Inc.) was used to validate MS/MS-based peptide and protein identifications. Stringency threshold parameters were set to 99.0% protein, minimum one peptide, and 95% peptide, leading to a list of 49 hits (including NGN3) identified in both HA-NGN3 immunoprecipitated samples, but not in the pcDNA3 control sample. After the exclusion of NGN3, the remaining 48 hits underwent Gene Onthology (GO) enrichment analysis. The top ten enriched GO molecular pathway terms were determined based on the p-value score of each term, as calculated by the EnrichR online resource[54,55]. The mass spectrometry proteomics data have been deposited to the ProteomeXchange Consortium via the PRIDE partner repository with the dataset identifier PXD033691 and 10.6019/PXD033691.

## Statistical analysis

Results are represented as mean ± standard error mean (SEM) for all bar graphs. Statistical significance was determined as described in the figure legends and calculated using GraphPad Prism 9. For all experiments comparing maximum two groups, p-values were calculated using an unpaired, two-tailed Student's t-test. Alternatively, a one- or two-way analysis of variance followed by Sidak's, Dunnett's, or Tukey's correction for multiple comparisons was used. A p value of < 0.05 was regarded as statistically significant for all data sets. P values are provided in the figure panels. The exact sample sizes (n) used to calculate statistics are provided in the figure legends. All experiments were reproduced with similar results a minimum of three times, unless otherwise specified. All immunoblots and micrograph images are representative of a minimum of three biologically independent experiments.

## Reporting summary

Further information on research design is available in the Nature Portfolio Reporting Summary linked to this article.

## Data availability

The mass spectrometry proteomics data shown in Supplementary Fig. 1 are available via ProteomeXchange with the identifier PXD033691. Additionally, the publicly available dataset used for Fig. 5 has been obtained and used in this manuscript (OMIX236[34]). Source data are provided with this paper.

## Code availability

Custom code used for the analysis of NGS data was written in R and is available from GitHub (https://github.com/SanchoLab/scRNAseq_Human_Foetal_Pancreas).

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

## Acknowledgements

We thank the Crick STPs and the King's College London Centre of Excellence for Mass Spectrometry for technical help and support. We thank Francesca Spagnoli and Clare Davies for the critical reading of the manuscript. T.M. was supported by a Wellcome Trust PhD fellowship (215138/Z/18/Z). C.M.G. was supported by a Wellcome Trust PhD fellowship (218461/Z/19/Z). J.K.N. and A.B. were supported by the Cancer Research UK (FC001039), the UK Medical Research Council (FC001039), the Wellcome Trust (FC001039), the Engineering and Physical Sciences Research Council (EP/T003103/1), and Breast Cancer Now (CTR-Q5-Y2). I.E. was supported by Cancer Research UK (FC001039) and Breast Cancer Now (CTR-Q5-Y2). K.H. was supported by the Swedish Research Council (2021-00201) and Breast Cancer Now (CTR-Q5-Y2). R.S. was supported by the MRC grant (MR/S000011/1) and MRC/JDRF grant (MR/T015470/1) to R.S.

## Author contributions

Conceptualization: T.M., J.K.N., and R.S. Formal analysis: T.M., C.M.G., J.K.N., and R.S. Investigation: T.M., C.M.G., K.H., I.E., and J.K.N. Writing (original draft): T.M., J.K.N., and R.S. Writing (review and editing): T.M, J.K.N, C.M.G., I.E., and R.S. Funding acquisition: A.B. and R.S. Supervision: R.S.

## Competing interests

The authors declare no competing interest.
