## [Peer Review File · Nature Communications]

USP7 controls NGN3 stability and pancreatic endocrine lineage developmentREVIEWER COMMENTS

Reviewer #1 (Remarks to the Author):

In this work the authors identified an interesting post-translation mechanism involved in pancreatic and endocrine development. More specifically, they found that the interaction of USP7 deubiquitinase with NGN3 (key regulator for endocrine specification), is crucial for NGN3 stabilization by deubiquitinating NGN3 and thus restraining it from entering proteasomal degradation during endocrine specification. In general, the study is very interesting and novel to its topic as it focuses in a different NGN3 regulatory level from the well explored transcriptional one. Yet, I do have the following comments before considering this work for publication.

Major points:

1) In general, the authors need to explain their data more clearly. Sometimes it is too abstract the way they mentioned their experimental setting and their findings. Also, there are several strong statements without much supportive data. I have specified some in my comments but the author need to check the manuscript thoroughly to adjust the tone.

2) "Pdx1-Cre; Usp7F/F mice also presented significantly smaller pancreases" which is shown in Fig. 3b. and "while the acinar and ductal compartments in the pancreas appeared normal..." This data is very surprising. It is well known that around 95% of pancreas is exocrine tissue and less than 5% is endocrine cells. If the mutant has no effect on exocrine tissue how it would result in more than 50% reduction in pancreas weight? In my opinion this massive reduction is likely not due to reduced islet cell area.

3) Page 7; "All together, these data suggest that Usp7 deletion leads to a significant reduction in NGN3 protein levels.." in my opinion this data show reduced number of Ngn3+ cells. It might imply reduced protein levels indirectly but the current conclusion is too strong. To confirm that USP7 loss result in reduced Ngn3 stability (which is the major claim of the study), the authors need to check the expression of Ngn3 mRNA in sorted mouse pancreatic epithelial cells. This will likely show the comparable amount of mRNA but different number of immunostained Ngn3+ cells. This is a key experiment to support the main claim of the study.

4) Figure 4 a-d is probably the key stage to confirm the main finding of the study. But this section is poorly presented. i) can the author check the exocrine compartment at this stage. Considering the huge loss of pancreas weight at adult stage this analysis is very important to be done using proper markers and quantification. ii) Some more analysis other than Ngn3 staining is needed. For instance, staining of other endocrine markers such as Nkx2-2 or ChgA will support this section. iii) Ngn3 is not only drives endocrine cells differentiation but coordinate their morphogenesis too. Can the author provide some staining showing the morphology of proto-islets at embryonic stages?

5) Page 4; "Moreover, this stabilization of NGN3 resulted from decreased NGN3 polyubiquitination and proteasomal degradation, which was dependent on USP7 catalytic activity (Figure 2a-d)." The figure show polyubiquitination but I can not get how they show degradation of Ngn3. Could the authors provide direct data showing less degradation of NGN3 in the presence of USP7?

6) On the stem cell differentiation system, treated the cells at different stages might be important and informative. Since USP7 is broadly expressed in different cell types, treatment at pancreatic progenitor stage might reveal if loss of USP7 function also impact the exocrine compartment or the progenitor number. Also, treatment at later stages or for longer time might make it clearer if loss of this protein impact alpha and delta cell number too. Additionally, can the overexpression of USP7 in endocrine progenitor increase Ngn3+ cells and eventually the number of hormone+ cells?

7) The discussion section is short and lack of deep discussing the finding of this study well into the

general direction of the field. The strength and limitation of the study need to be discussed well.

Other points:

8) It is important to mention which model system has been used for the overexpression studies at least once in the result section.

9) Page 5; "To this end, we generated pancreas-specific conditional Usp7 knockout mice." Is the flox mice generated in this study or the authors used already available mouse line. This info needs to be mentioned in the result section. And is the Cre driver constitutive?

10) Page 6; "while the acinar and ductal compartments in the pancreas appeared normal..." How they measure this?

11) Page 6; "Taken together, these findings suggest that USP7 plays a key role in pancreatic endocrine development and function." Loss of USP7 showed reduction in islet cell area. There is no data to show that it impacts islet cell function.

12) Page 7; "This was particularly evident in the PDX1^{high} fraction, consisting of beta cells and branching epithelial cells (Fig. 4c)." It is not clear what the authors refer to.

13) There are very efficient protocols in place for iPSC to beta cell differentiation, maybe authors could explain if there is a benefit to utilize the protocol implemented in this study. From the Immunofluorescent images, the amount of insulin positive cells in the untreated condition is not very high from what you would observe with other protocols. Maybe the authors could use a better picture that reflects the quantifications.

14) Maybe the authors could include a couple more current publications/studies that have also encounter key post-translational regulators during pancreatic development that might complement the discussion if relevant to NGN3.

15) Maybe authors could elaborate or discuss about other mechanisms that might be compensating in order to create some NGN3 cells even without USP7 post-translational regulator.

16) Fig. 1e; Consider adding the values for USP7 into the graph.

17) Fig. 2a; Isn't increased poly-Ub levels result in decreased Ngn3 protein levels. This can not be seen in the first and second lanes (from left) in the input fraction. I would expect less protein in the Ub-His sample compared to the HA-NGN3 only expressing cells. The same applies to Fig 2c, where one expects less protein levels in the input of cells expressing USPC223A compared to those expressing USP7WT.

18) Fig. 3d what it means number of islets? How the author quantified this? The proper way of doing this would be to measure total islet number in the whole pancreas and/or measure islet cell mass per pancreas. Islets are not homogenously scattered within the pancreas. So, number of islets per specific region of pancreas is not informative.

19) Fig. 3f, h. again it is not clear how this quantification has been done. For hormone+ cells, cell mass over whole pancreatic weight is the proper way to doing this.

20) Fig. 4c. A epithelial/ductal marker such as F-actin would make the staining clearer.

21) Fig. 4g; why the cell co-expressing NGN3 and USP7 are low. Form the major conclusion of this study, one would expect to see expression of USP7 if not in all but in a high percentage of endocrine progenitors. It is also mentioned that "...USP7 is reliably expressed during the NEUROG3 expression

window (Fig. 4h), peaking at W14 and preceding the NEUROG3 expression peak (W16).” Then why these two genes do not high overlap in endocrine progenitors?

22) Fig. 4g. It would be better to have a zoom in the endocrine part. Ideally showing USP7 expression in different cell types using violin plots or dot plots would be more informative than the UMAP.

23) Supp Fig. 3. I was wondering if it would be possible to have USP7 staining in WT and CKO by confocal microscopy.

Reviewer #2 (Remarks to the Author):

The current manuscript by Manea et al. utilized mass spectroscopy to identify regulators of NGN3. Using this approach, they found USP7 to directly interact with NGN3 and further studied its importance in a mouse conditional knockout model. Pancreas-specific knockout of USP7 resulted in a significant reduction of pancreas size and the number of islets in adult mice. Utilizing a human induced pluripotent stem cell differentiation, the authors also tested the effects of inhibition of USP7 on NGN3 expression in endocrine progenitors and INS+ beta-like cells. Overall, the current study demonstrates the role of USP7 in stabilizing NGN3 to promote endocrine specification. Overall, this manuscript is well written and provides new insights into pancreas endocrine specification via post-translational regulation (on NGN3).

This manuscript should be further revised to provide more mechanistic evidence of how USP7 can be relevant in human disease or islet function. At minimum, the following concerns should be addressed before the manuscript is accepted for publication.

For the Figure 1 and 2, authors could consider including more details for each graph and conclusion. For readers it is easy to get lost in figure 1 & 2 plus all supplementary figures. For example, include in the labeling or clearly state in the legend about what cells are using for the experiments.

-Figure 3d, 3f, 3h – The authors need to specify how the quantification was completed. Are the percentages based on total pancreas area? Pancreas sections? It needs to be consistent. In addition, based on the Figure 3C, Authors mention a decrease in the number of islets and probably in islet size. Did authors analyze proliferation in early stages?

-Figure 4c – In the main text, lines 169-173, the authors mention the number of PDX1+ cells was unaffected (data not shown), however the expression level appeared less in the Pdx1-Cre; Usp7F/F. “This was particularly evident in the PDX1high fraction, consisting of beta cells and branching epithelial cells”. Due to the clarity of the images and low levels of Pdx1, it is difficult to determine the number of PDX1+ cells. In addition, when looking at the PDX1high fraction it is difficult to identify the beta cells, the branching epithelial cells, and the NGN3+ cells. It appears that there are no NGN3+ cells in the Pdx1-Cre; Usp7F/F pancreas. The authors should revise the statements or provide an image that is representative of the quantification data.

-Have the authors checked NGN3 expression at other time points such as E10.5 or E12.5 to see if the reduction in NGN3 expression is observed at other developmental time points?

-Figure 4h – The authors state that, “...bubble expression plots...demonstrated that USP7 is reliably

expressed during the NEUROG3 expression peak (W16).” It is unclear if the data shows this clear expression of both USP7 and NEUROG3.

-Figure 5 – The iPSC differentiation model seems very inefficient for producing endocrine cells. The percentage of NGN3+ cells is very low. Did the authors try alternative differentiation protocols to improve NGN3 expression and determine the effects from USP7 inhibition?

Response to reviewer comments for NCOMMS-22-20842-T 'USP7 controls NGN3 stability and pancreatic endocrine lineage development' by Manea et al.

We are pleased that the reviewers recognised the importance of our work on the post-translational regulation of Ngn3 by USP7, and the relevance for the field. We would like to thank the reviewers for their critical assessment and constructive comments, which have greatly helped us to improve our study. To address the concerns raised by the reviewers, we have generated a substantial amount of new data that we hope clarify all the reviewers' criticisms and suggestions as detailed in the point-by-point response below. The original referee comments are in *italic black text*, and our point-by-point response is in **blue text** below.

REVIEWER 1

In this work the authors identified an interesting post-translation mechanism involved in pancreatic and endocrine development. More specifically, they found that the interaction of USP7 deubiquitinase with NGN3 (key regulator for endocrine specification), is crucial for NGN3 stabilization by deubiquitinating NGN3 and thus restraining it from entering proteasomal degradation during endocrine specification. In general, the study is very interesting and novel to its topic as it focuses in a different NGN3 regulatory level from the well explored transcriptional one. Yet, I do have the following comments before considering this work for publication.

Major points:

1) In general, the authors need to explain their data more clearly. Sometimes it is too abstract the way they mentioned their experimental setting and their findings. Also, there are several strong statements without much supportive data. I have specified some in my comments but the author need to check the manuscript thoroughly to adjust the tone.

We apologise if the reviewer felt that the data was not clearly explained. Following the reviewer's suggestion, we have now reworded the manuscript thoroughly to make it clearer and to adjust the tone.

2) "Pdx1-Cre; USP7^{F/F} mice also presented significantly smaller pancreases" which is shown in Fig.3b. and "while the acinar and ductal compartments in the pancreas appeared normal..." This data is very surprising. It is well known that around 95% of pancreas is exocrine tissue and less than 5% is endocrine cells. If the mutant has no effect on exocrine tissue how it would result in more than 50% reduction in pancreas weight? In my opinion this massive reduction is likely not due to reduced islet cell area.

We thank the reviewer for this important comment. We agree that the original representation of the absolute pancreas weight was misleading and representing the ratio pancreas weight/total weight is more accurate. The data is now displayed as the ratio pancreas weight/total weight of the mouse (new Fig.3b). We find no significant differences in the ratio of pancreas/body weight between the different genotypes. Our new data (new Fig.3b) indicates that the previous reduction in absolute pancreas weight was due to an overall reduction in body weight and size. We have now addressed this point in the revised discussion where we hypothesise that the reduced number of insulin-producing beta cells in *Pdx1-Cre; USP7^{F/F}* causes subsequent growth restrictions, as it has been previously reported in *INS*-deficient mice (Duvillié et al., 1997).

*3) Page 7; "All together, these data suggest that *Usp7* deletion leads to a significant reduction in NGN3 protein levels.." in my opinion this data show reduced number of Ngn3+*

cells. It might imply reduced protein levels indirectly but the current conclusion is too strong. To confirm that USP7 loss result in reduced Ngn3 stability (which is the major claim of the study), the authors need to check the expression of Ngn3 mRNA in sorted mouse pancreatic epithelial cells. This will likely show the comparable amount of mRNA but different number of immunostained Ngn3+ cells. This is a key experiment to support the main claim of the study.

This is an important point, and the reviewer is right that we can only claim a reduction of NGN3+ cells upon *Usp7* deletion from the data presented in the submitted Fig.4. We have now reworded this in the revised version of the manuscript.

The reviewer suggested that sorting mouse epithelial cells and checking the mRNA expression of *Neurog3* would be a better way of addressing if USP7 controls NGN3 stability at the post-translational level in the mouse embryonic pancreas. Although this is a good strategy, endogenous *Neurog3* expression occurs only in the embryonic pancreatic epithelial cells, which are technically challenging to sort due to limited material. Moreover, the interpretation of that experiment would be compromised as NGN3 controls its own transcriptional expression (Ejarque et al., 2013; Shih et al., 2012), therefore it would be difficult to uncouple NGN3 protein levels and *Neurog3* mRNA expression. Nevertheless, we have performed RNAscope to visualise *Neurog3* mRNA at E14.5 (Supplementary Fig.5a) and found that *Neurog3* mRNA expression is not significantly changed in *Pdx1-Cre; USP7^{F/F}* E14.5 embryos (Supplementary Fig.5a). Furthermore our *in vitro* biochemical data shows that USP7 functions as a deubiquitinating enzyme to stabilize NGN3 protein (Fig.1,2 and Supplementary Fig.1,2). Specifically, when *Neurog3* expression is controlled ectopically through a stable promoter, USP7 regulates NGN3 protein levels in a post-translational manner. Taken together these data suggest that USP7 regulates NGN3 stability, however we cannot exclude that additional mechanisms operate to control NGN3 activity. We have included this point in the new discussion.

4) Figure 4 a-d is probably the key stage to confirm the main finding of the study. But this section is poorly presented. i) can the author check the exocrine compartment at this stage. Considering the huge loss of pancreas weight at adult stage this analysis is very important to be done using proper markers and quantification. ii) Some more analysis other than Ngn3 staining is needed. For instance, staining of other endocrine markers such as Nkx2-2 or ChgA will support this section. iii) Ngn3 is not only drives endocrine cells differentiation but coordinate their morphogenesis too. Can the author provide some staining showing the morphology of proto-islets at embryonic stages?

We agree with the reviewer that further analysis of the embryonic pancreas of *Usp7* knock-out would strengthen our study. We have performed a comprehensive analysis to determine the effect of *Usp7* loss on the exocrine compartment, progenitors and proto-islets. To address this point, we have used AMY (acinar marker), OPN (ductal marker at this stage of embryonic development), NKX2.2 and CHGA (endocrine markers), as well as Insulin (to detect proto-islets) in *Usp7^{F/F}* and *Pdx1-Cre;Usp7^{F/F}* mouse E14.5 embryo sections. While we found no significant differences in the proportion of AMY+ acinar cells or OPN+ ductal cells (New Supplementary Fig.5b) between genotypes, *Pdx1-Cre;Usp7^{F/F}* embryos showed significantly decreased percentages of NKX2.2+ cells (New Fig.4e-f), CHGA+ cells (New Fig.4g-h) and INS+ cells (New Fig.4g,i). And while proto-islets were frequent in E14.5 *Usp7^{F/F}* embryos, they were rare and smaller in the *Pdx1-Cre; USP7^{F/F}* (New Fig.4g-low mag). The reduction of NKX2.2, CHGA, INS+ and proto-islets together with the acinar and ductal compartment not being affected is consistent with the reduction of NGN3+ cells observed in the *Pdx1-Cre; USP7^{F/F}* mice and with the Ngn3 knock-out phenotype previously reported (Gradwohl et al., 2000).

5) Page 4; “Moreover, this stabilization of NGN3 resulted from decreased NGN3 polyubiquitination and proteasomal degradation, which was dependent on USP7 catalytic activity (Figure 2a-d).” The figure show polyubiquitination but I can not get how they show degradation of Ngn3. Could the authors provide direct data showing less degradation of NGN3 in the presence of USP7?

We apologise if this point was not made clear in our manuscript. The ubiquitination pull-down experiments were performed in the presence of the proteasomal inhibitor MG132 which leads to stabilization and accumulation of ubiquitinated NGN3 (Fig.2a-d). Importantly, this ubiquitinated pool of NGN3 was sensitive to wild-type USP7 and not a catalytic mutant USP7 (USP7^{C223A}). We have now clarified these findings in the results section.

Direct evidence for decreased degradation of NGN3 upon USP7 overexpression is shown in the cycloheximide pulse-chase experiments in Fig.1f-g from our original version of the manuscript (New Fig.1f-g). Overexpression of USP7 wild-type leads to a clear increase in NGN3 stabilisation (as measured by NGN3 half-life), while overexpression of catalytically inactive USP7 (USP7^{C223A}) had no effect on NGN3 stability (New Fig.1f-g). In these experiments NGN3 expression is controlled through an ectopic constitutive promoter (CMV), and protein synthesis is inhibited via cycloheximide treatment. The observed increase in NGN3 protein half-life upon USP7 overexpression is therefore independent of newly transcribed NGN3 mRNA, but due to stabilization of NGN3.

6) On the stem cell differentiation system, treated the cells at different stages might be important and informative. Since USP7 is broadly expressed in different cell types, treatment at pancreatic progenitor stage might reveal if loss of USP7 function also impact the exocrine compartment or the progenitor number. Also, treatment at later stages or for longer time might make it clearer if loss of this protein impact alpha and delta cell number too. Additionally, can the overexpression of USP7 in endocrine progenitor increase Ngn3+ cells and eventually the number of hormone+ cells?

We thank the reviewer for this important comment, and we agree that inhibiting USP7 at different stages of beta cell differentiation may offer more information about the role of USP7 in the development of different pancreatic cell types. To address this point, alongside other reviewer suggestions about the use of alternative differentiation protocols (See reviewer 2, last point), we repeated the differentiation experiments on 3D pancreatic progenitor domes using a protocol from Russ and colleagues (Russ et al., 2015) with modifications implemented by Trott and colleagues (Trott et al., 2017) (New Fig.7).

Following the reviewer suggestion, we treated the cells with the USP7 inhibitor (GNE-6640) at different stages: Pancreas progenitors (PP) (Day 0 - Day5), Endocrine progenitor (EP) stage (Day 5- Day 9), Endocrine Maturation stage (Day 9-Day 16) and Pancreas progenitor (PP) + Endocrine progenitor (EP) stages (Day 0 - Day 9) (New Fig.7a) and measured the effect on Endocrine progenitors (NGN3+) and Endocrine cells (INS+ or SST+) differentiation.

The new data, presented in Fig.7, demonstrates that inhibiting USP7 during the PP stage (Day 0-5) has no effect on the % of NGN3+ cells by the end of the endocrine induction stage. However, when cells are treated with the USP7 inhibitor during the EP stage (Day 5-9) or during both PP & EP stages (Day 0-9) the % of NGN3+ cells is significantly reduced. Accordingly, the % of INS+ cells at the end of the endocrine maturation stage was not affected when cells were treated with the USP7 inhibitor during the PP stage (Day 0-5) or during the Endocrine maturation stage (Day 9-16), but a clear decrease in the % of INS+ cells

was observed when cells were treated with the USP7 inhibitor during the EP stage (Day 5-9) (New Fig.7d-e).

These results indicate that USP7 inhibition at the pancreatic progenitor stage, when NGN3 is still not present, does not impact generation of NGN3+ endocrine progenitors or INS+ cells. Conversely, inhibiting USP7 in stages where NGN3 is present (EP) results in a clear reduction of Endocrine progenitors and INS+ cells (New Fig.7).

Regardless of the timing and length of USP7 inhibition, we find no significant reduction in the percentage of SST+ delta cells at the end of the differentiation (New Fig.7d-e). Very few GCG+ cells could be observed in both the control and treated conditions (<1%) when using this differentiation protocol. Acinar and ductal cell populations are not generally present at the end of the Russ or Hoglebe differentiation protocols, as these protocols aim to direct pancreatic progenitors towards endocrine cell fates. Therefore, these assays would not allow us to further investigate the effect of USP7 on acinar and ductal populations *in vitro*.

We agree that overexpressing USP7 during iPSC-to-Beta-cell differentiation could provide additional insight into NGN3 regulation and beta cell generation. Following the reviewer suggestion, we attempted to generate stable iPSCs and PP organoids overexpressing USP7 by infecting them with the lentiviral vector pLV-Flag-USP7-EGFP (Rebuttal Figure 1). However, we could not detect any significant change in USP7 gene expression by qPCR analysis between pLV-mCherry-EGFP and pLV-FlagUSP7 infected cells (Rebuttal Figure 1), which could be consistent with a deleterious effect of sustained overexpression of USP7 in iPSC or PP organoids. We therefore could not generate an appropriate *in vitro* model to study the effect of USP7 overexpression on endocrine cell generation. While one plausible alternative would be to generate a Dox inducible USP7 line, due to the time limitations we could not implement that strategy.

7) The discussion section is short and lack of deep discussing the finding of this study well into the general direction of the filed. The strength and limitation of the study need to be discussed well.

We have now extended our discussion with special focus on the findings of this study and the direction in the field and discussed the strengths and limitations of the study.

Other points:

8) It is important to mention which model system has been used for the overexpression studies at least once in the result section.

We apologise for this oversight and have now clarified this point in the results and methods section.

9) Page 5; "To this end, we generated pancreas-specific conditional Usp7 knockout mice." Is the flox mice generated in this study or the authors used already available mouse line. This info needs to be mentioned in the result section. And is the Cre driver constitutive?

We apologise for this oversight and have now clarified throughout the manuscript. The *Usp7-Flox* mouse was previously generated and described (Kon et al., 2011). The *Usp7-Flox* mice were crossed to *Pdx1-Cre* mice (Hingorani et al., 2003) to generate the pancreas specific conditional *Usp7* knock-out mice in our study. In the *Pdx1-Cre* mouse line, the expression of Cre is not inducible, but it is constitutively expressed in *Pdx1* expressing cells, which are only present in the embryonic stage of pancreas development. *Pdx1* is an established pancreatic-specific Cre driver of expression.

10) Page 6; “while the acinar and ductal compartments in the pancreas appeared normal...”
How they measure this?

This was assessed histologically by H&E and by quantification of the % of Acinar and Ductal cells after immunofluorescent staining of adult *Usp7^{F/F}* and *Pdx1-Cre; Usp7^{F/F}*; adult pancreata with specific cellular markers for acinar (amylase) and ductal (Ck19) cells. We observed no difference in presentation or morphology of either cellular compartment, nor significant differences in expression of specific markers for acinar (amylase) and ductal (Ck19) cell types between the different genotypes (New Fig.3e-f and New Supplementary Fig.3a).

11) Page 6; “Taken together, these findings suggest that USP7 plays a key role in pancreatic endocrine development and function.” Loss of USP7 showed reduction in islet cell area. There is no data to show that it impacts islet cell function.

We assessed islet cell function (indirectly) through its ability to regulate circulating blood glucose levels via secretion of insulin (New Fig.3b). Due to impairment of Islet number, size and insulin expression, we found that *Pdx1-Cre; Usp7^{F/F}* mice had significantly higher circulating blood glucose levels, compared to *Usp7^{F/F}* control mice. This point has now been clarified in the result section.

12) Page 7; “This was particularly evident in the PDX1^{high} fraction, consisting of beta cells and branching epithelial cells (Fig.4c).” It is not clear what the authors refer to.

As described in previous studies (Wescott et al., 2009) we observe differential PDX1 expression within the mouse embryonic pancreas cells, with distinct PDX1^{low} and PDX1^{high} cell populations. The PDX1^{high} population mainly consists of beta cells, co-staining with INS (New Fig.4g). *Pdx1-Cre; Usp7^{F/F}* E14.5 embryos show a reduced proportion of INS⁺ cells compared to control, likely explaining why the PDX1^{high} population is almost absent from the pancreas of E14.5 *Pdx1-Cre; Usp7^{F/F}* embryos. This point has now been clarified in the new version of the manuscript.

13) There are very efficient protocols in place for iPSC to beta cell differentiation, maybe authors could explain if there is a benefit to utilize the protocol implemented in this study. From the Immunofluorescent images, the amount of insulin positive cells in the untreated condition is not very high from what you would observe with other protocols. Maybe the authors could use a better picture that reflects the quantifications.

This point has now been discussed in our revised discussion. We have previously obtained higher differentiation efficiency (25%[<] INS⁺ cells) using the Russ protocol (Russ et al., 2015) by differentiating pancreatic progenitors as aggregates in Aggrewell plates without Matrigel embedding. While using cells in suspension protocols is good, we reasoned that providing the 3D environment is more relevant to the *in vivo* physical cues the cells are exposed to, reaching % of NGN3⁺ cells similar to those observed during embryonic development (New Fig.4 and new supplementary Fig.4). Additionally, it allows consistent measures for cell types thereby reducing the variability. Though the efficiency of differentiation is slightly reduced in our 3D differentiation protocol (Cujba et al., 2022; Pedraza-Arevalo et al., 2022), it does allow comparison between conditions in a setting where the cells are receiving physical cues more similar to the *in vivo* state than when cells are grown in suspension.

We have revised the IF images and used those that are more representative for the quantifications (New Fig.6d, high mag and New Fig.7d).

14) Maybe the authors could include a couple more current publications/studies that have also encounter key post-translational regulators during pancreatic development that might

complement the discussion if relevant to NGN3.

We have extended our discussion to incorporate these suggestions.

15) *Maybe authors could elaborate or discuss about other mechanisms that might be compensating in order to create some NGN3 cells even without USP7 post-translational regulator.*

Following the reviewer suggestion, we have now included a section in the discussion focusing on the possible mechanisms that might be compensating in order to create some rare NGN3 cells in the *Pdx1-Cre; Usp7^{F/F}*. We discuss the possibility of those cells being escapers of the Cre mediated deletion and the existence of other DUBs in some NGN3+ progenitors.

16) *Fig. 1e; Consider adding the values for USP7 into the graph.*

We have now included the values for USP7 into our new Supplementary Fig.2a.

17) *Fig. 2a; Isn't increased poly-Ub levels result in decreased Ngn3 protein levels. This can not be seen in the first and second lanes (from left) in the input fraction. I would expect less protein in the Ub-His sample compared to the HA-NGN3 only expressing cells. The same applies to Fig 2c, where one expects less protein levels in the input of cells expressing USPC223A compared to those expressing USP7WT.*

It is important to note that these experiments were conducted in the presence of a proteasome inhibitor (MG132), which allows for stabilization and accumulation ubiquitinated NGN3 protein. Hence, why we see each sample containing similar levels of NGN3 protein, despite differences in ubiquitination status. Moreover, our results show that this accumulation of ubiquitinated NGN3 species is sensitive to USP7 deubiquitination and requires its catalytic activity. This has now been better clarified in the result section.

18) *Fig. 3d what it means number of islets? How the author quantified this? The proper way of doing this would be to measure total islet number in the whole pancreas and/or measure islet cell mass per pancreas. Islets are not homogenously scattered within the pancreas. So, number of islets per specific region of pancreas is not informative.*

In Fig. 3D, the total islet number was measured throughout the whole pancreas section, not in a specific region. Moreover, the value is averaged from multiple (>3) section levels of the whole pancreas, as indeed islets are not homogenously scattered within the pancreas. Details on quantifications are described fully in Methods section.

19) *Fig. 3f, h. again it is not clear how this quantification has been done. For hormone+ cells, cell mass over whole pancreatic weight is the proper way to doing this.*

We apologise if this was not clear enough in the Methods sections. In Fig. 3 all the quantifications shown are the percentages of the area with positive cell staining relative to the total pancreas area (% Area), not a specific section or region. The % Area values for each cell type from the multiple section levels of whole pancreas tissue (minimum 3 technical samples) are averaged, and each data point represented in the graph corresponds to the average % Area of a specific cell type for each mouse's pancreas analysed.

20) *Fig. 4c. A epithelial/ductal marker such as F-actin would make the staining clearer.*

We have selected a more representative picture for the previous Fig. 4c. Furthermore, in order to analyse the effect of *Usp7* knock-out on the ductal compartment of the pancreas,

we have now carried out immunofluorescent staining of the E14.5 embryo sections with ductal marker osteopontin (OPN), providing a clearer image of the pancreatic duct structure (New Supplementary Fig.5b).

21) Fig.4g; why the cell co-expressing NGN3 and USP7 are low. Form the major conclusion of this study, one would expect to see expression of USP7 if not in all but in a high percentage of endocrine progenitors. It is also mentioned that “...USP7 is reliably expressed during the *NEUROG3* expression window (Fig.4h), peaking at W14 and preceding the *NEUROG3* expression peak (W16).” Then why these two genes do not high overlap in endocrine progenitors?

This is a very interesting point. To explore this further we have performed a deeper analysis of the RNAseq dataset from human pancreas. As the reviewer pointed out, the scRNAseq data shows that *USP7* is expressed in a subset of *NEUROG3* expressing cells (New Fig.5c,f,g-i). While this could be counterintuitive at first, it could just indicate that only certain NGN3+ cells are regulated by *USP7*, which is consistent with the iPSC-differentiation results where we still observe NGN3+ cells at the end of the endocrine progenitor stage in cells treated with the *USP7* inhibitor GNE-6640. Interestingly, we observed that the percentage of *USP7* expressing cells varies during the different gestational weeks in *NEUROG3*+ cells but not in *NEUROG3*- cells (New Fig.5h). While the % of *USP7* expressing cells is maintained constant in *NEUROG3*- cells across different gestational weeks, the percentage of *USP7* expressing cells in *NEUROG3*+ cells increases gradually until reaching more than 30% at gestational week 14, coinciding with the beta cell generation window of human beta cells (Piper et al., 2004). This is consistent with the notion that *USP7* regulation at that stage plays an important role for beta cell development. We have now included this in the revised version of the manuscript.

22) Fig.4g. It would be better to have a zoom in the endocrine part. Ideally showing *USP7* expression in different cell types using violin plots or dot plots would be more informative than the UMAP.

Following the reviewer suggestion, we have explored the expression of *USP7* and *NEUROG3* within the endocrine part. To do so, we generated an object containing EP, alpha, beta, delta and PP-cells and represented the *NEUROG3/USP7* co-expression UMAP and expression dot plot with the different cell types in it (New Fig.5e-g). As depicted in the dot plot (New Fig.5g) we observed *USP7* expression in a subset of EP and beta cells, with the levels of *USP7* expression being the highest in the EP cells. This has now been included in the revised version of the manuscript.

23) Supp Fig.3. I was wondering if it would be possible to have *USP7* staining in WT and CKO by confocal microscopy.

Unfortunately, the *USP7* antibodies we have trialled (Abcam: ab4080, Bethyl Laboratories: A300-033A, Bethyl Laboratories: A300-034A) have not been specific for *USP7* detection via immunofluorescent staining in mouse pancreas sections due to their overall background/unspecific signal. That was in part the reason why we included *USP7* IHC in Fig.3, where specific staining for *USP7* can be appreciated in the nucleus of islet cells and scattered ductal cells in control mice, while the signal is virtually absent in the *USP7* knock-out mice.

Reviewer #2 (Remarks to the Author):

The current manuscript by Manea et al. utilized mass spectroscopy to identify regulators of NGN3. Using this approach, they found USP7 to directly interact with NGN3 and further studied its importance in a mouse conditional knockout model. Pancreas-specific knockout

of USP7 resulted in a significant reduction of pancreas size and the number of islets in adult mice. Utilizing a human induced pluripotent stem cell differentiation, the authors also tested the effects of inhibition of USP7 on NGN3 expression in endocrine progenitors and INS⁺ beta-like cells. Overall, the current study demonstrates the role of USP7 in stabilizing NGN3 to promote endocrine specification. Overall, this manuscript is well written and provides new insights into pancreas endocrine specification via post-translational regulation (on NGN3). This manuscript should be further revised to provide more mechanistic evidence of how USP7 can be relevant in human disease or islet function. At minimum, the following concerns should be addressed before the manuscript is accepted for publication.

For the Figure 1 and 2, authors could consider including more details for each graph and conclusion. For readers it is easy to get lost in figure 1 & 2 plus all supplementary figures. For example, include in the labeling or clearly state in the legend about what cells are using for the experiments.

We apologise if the reviewer felt that this was not clear. We have now included more details for each graph and conclusion and indicated in the figure legends the cells used for each experiment.

-Figure 3d, 3f, 3h - The authors need to specify how the quantification was completed. Are the percentages based on total pancreas area? Pancreas sections? It needs to be consistent. In addition, based on the Figure 3C, Authors mention a decrease in the number of islets and probably in islet size. Did authors analyze proliferation in early stages?

We apologise if this was not clear enough in the Methods sections. In Fig.3 all the quantifications shown are the percentages of the area with positive cell staining relative to the total pancreas area (% Area). The % Area values for each cell type from the multiple section levels of whole pancreas tissue (minimum 3 technical samples) are averaged, and each data point represented in the graph corresponds to the average % Area of a specific cell type for each mouse's pancreas analysed.

Following the reviewer's suggestion, we analysed the proliferation marker Ki67 in early stages (E12.5) (New Supplementary Fig.4c) and detected no significant difference in the % of Ki67 cells between genotypes. Consistent with a *Ngn3* knock-out phenotype (Gradwohl et al., 2000), we reasoned the decrease in islet number and size is due to the reduction in *Ngn3* observed at E14.5 (when the endocrine cells are specified) in the *Pdx1-Cre; Usp7^{F/F}* mice.

-Figure 4c - In the main text, lines 169-173, the authors mention the number of PDX1⁺ cells was unaffected (data not shown), however the expression level appeared less in the *Pdx1-Cre; USP7^{F/F}*. "This was particularly evident in the PDX1^{high} fraction, consisting of beta cells and branching epithelial cells". Due to the clarity of the images and low levels of *Pdx1*, it is difficult to determine the number of PDX1⁺ cells. In addition, when looking at the PDX1^{high} fraction it is difficult to identify the beta cells, the branching epithelial cells, and the NGN3⁺ cells. It appears that there are no NGN3⁺ cells in the *Pdx1-Cre; USP7^{F/F}* pancreas. The authors should revise the statements or provide an image that is representative of the quantification data.

We apologise if the reviewer felt this was not clear. We have now reworded the text to incorporate the reviewer suggestion and provided images that are representative of the quantifications and better depict the Pdx1⁺ cells (New Fig.4c,e,g). As described in previous studies (Wescott et al., 2009) we observe differential PDX1 expression within the mouse embryonic pancreas cells, with distinct PDX1^{low} and PDX1^{high} cell populations. The PDX1^{high} population mainly consists of beta cells, co-staining with INS and CHGA (New Fig.4g, high

magnification). *Pdx1-Cre;Usp7^{F/F}* E14.5 embryos show a reduced proportion of INS⁺ cells compared to control, likely explaining why the PDX1^{high} population is almost absent from the pancreas of E14.5 *Pdx1-Cre;Usp7^{F/F}* embryos (New Fig.4g). This point has now been clarified in the new version of the manuscript.

-Have the authors checked NGN3 expression at other time points such as E10.5 or E12.5 to see if the reduction in NGN3 expression is observed at other developmental time points?

This is a very interesting point. Following the reviewer suggestion we have analysed NGN3 expression in E12.5 pancreata from *Usp7^{F/F}* and *Pdx1-Cre;Usp7^{F/F}* embryos (New Supplementary Fig.4b). At this timepoint, the number of NGN3⁺ cells in control mice (*Usp7^{F/F}*) is relatively small, and no significant difference was observed overall in the % of NGN3⁺ cells in the *Pdx1-Cre;Usp7^{F/F}* pancreas. One plausible explanation for this result is that USP7 activation or levels at E12.5 are still low to act on NGN3, therefore, due to the small number of cells expressing NGN3 at this stage it is difficult to capture any difference due to litter to litter variability.

-Figure 4h - The authors state that, "...bubble expression plots...demonstrated that USP7 is reliably expressed during the NEUROG3 expression peak (W16)." It is unclear if the data shows this clear expression of both USP7 and NEUROG3.

We apologise if this was not clear. To explore this further we have performed a deeper analysis of the RNAseq dataset from human pancreas. As per our answer to Reviewer 1 (Point 21) our scRNAseq data analysis shows that *USP7* is expressed in a subset of *NEUROG3* expressing cells (New Fig.5c,f,g-i). While this could be counterintuitive at first, it could just indicate that only certain *NEUROG3*⁺ cells are regulated by *USP7*, which is consistent with the iPSC-differentiation results where we still observe NGN3⁺ cells at the end of the endocrine progenitor stage in cells treated with the USP7 inhibitor GNE-6640. Interestingly, we observed that the percentage of *USP7* expressing cells varies during the different gestational weeks in *NEUROG3*⁺ cells but not in *NEUROG3*⁻ cells (New Fig.5h). While the % of *USP7* expressing cells is maintained at a constant level in *NEUROG3*⁻ cells across different gestational weeks, the percentage of *USP7* expressing cells in *NEUROG3*⁺ cells increases gradually until reaching more than 30% at gestational week 14, coinciding with the beta cell generation window of human beta cells (Piper et al., 2004). This is consistent with the notion that USP7 regulation at that stage plays an important role for beta cell development. We have now included this in the revised version of the manuscript.

-Figure 5 - The iPSC differentiation model seems very inefficient for producing endocrine cells. The percentage of NGN3+ cells is very low. Did the authors try alternative differentiation protocols to improve NGN3 expression and determine the effects from USP7 inhibition?

As per our response to Reviewer 1 (Point 13), this point has now been discussed in our revised manuscript. We have previously obtained higher differentiation efficiency (>25% INS⁺ cells) using the Russ protocol (Russ et al., 2015), by differentiating pancreatic progenitors as aggregates in Aggrewell plates without Matrigel embedding. While using cells in suspension protocols are good, we reasoned that providing the 3D environment is more relevant to the in vivo physical cues the cells are exposed to, reaching % of NGN3⁺ cells similar to those observed in during embryonic development (New Fig.4 and new supplementary Fig.4). Additionally, it allows consistent measures for cell types thereby reducing the variability. Although the efficiency of differentiation is slightly reduced in our 3D differentiation protocol (Cujba et al., 2022; Pedraza-Arevalo et al., 2022) it does allow comparison between

conditions in a setting where the cells are receiving physical cues more similar to the in vivo state than when cells are grown in suspension.

We have now carried out additional differentiation experiments on our 3D pancreatic progenitor domes using a protocol from Russ and colleagues (Russ et al., 2015) with modifications implemented by Trott and colleagues (Trott et al., 2017) (New Fig.7). However, the yield in NGN3 was similar to the ones obtained with our other protocol in New Fig.6.

References

- Cujba, A. M., Alvarez-Fallas, M. E., Pedraza-Arevalo, S., Laddach, A., Shepherd, M. H., Hattersley, A. T., Watt, F. M., & Sancho, R. (2022). An HNF1 α truncation associated with maturity-onset diabetes of the young impairs pancreatic progenitor differentiation by antagonizing HNF1B function. *Cell Reports*, 38(9), 110425. <https://doi.org/10.1016/J.CELREP.2022.110425>
- Duvillié, B., Cordonnier, N., Deltour, L., Dandoy-Dron, F., Itier, J. M., Monthieux, E., Jami, J., Joshi, R. L., & Bucchini, D. (1997). Phenotypic alterations in insulin-deficient mutant mice. *Proceedings of the National Academy of Sciences of the United States of America*, 94(10), 5137. <https://doi.org/10.1073/PNAS.94.10.5137>
- Ejarque, M., Cervantes, S., Pujadas, G., Tutusaus, A., Sanchez, L., & Gasa, R. (2013). Neurogenin3 Cooperates with Foxa2 to Autoactivate Its Own Expression. *The Journal of Biological Chemistry*, 288(17), 11705. <https://doi.org/10.1074/JBC.M112.388173>
- Gradwohl, G., Dierich, A., LeMeur, M., & Guillemot, F. (2000). neurogenin3 is required for the development of the four endocrine cell lineages of the pancreas. *Proceedings of the National Academy of Sciences of the United States of America*, 97(4), 1607-1611. <https://doi.org/10.1073/PNAS.97.4.1607>
- Hingorani, S. R., Petricoin, E. F., Maitra, A., Rajapakse, V., King, C., Jacobetz, M. A., Ross, S., Conrads, T. P., Veenstra, T. D., Hitt, B. A., Kawaguchi, Y., Johann, D., Liotta, L. A., Crawford, H. C., Putt, M. E., Jacks, T., Wright, C. V. E., Hruban, R. H., Lowy, A. M., & Tuveson, D. A. (2003). Preinvasive and invasive ductal pancreatic cancer and its early detection in the mouse. *Cancer Cell*, 4(6), 437-450. [https://doi.org/10.1016/S1535-6108\(03\)00309-X](https://doi.org/10.1016/S1535-6108(03)00309-X)
- Kon, N., Zhong, J., Kobayashi, Y., Li, M., Szabolcs, M., Ludwig, T., Canoll, P. D., & Gu, W. (2011). Roles of HAUSP-mediated p53 regulation in central nervous system development. *Cell Death & Differentiation* 2011 18:8, 18(8), 1366-1375. <https://doi.org/10.1038/cdd.2011.12>
- Pedraza-Arevalo, S., Cujba, A. M., Alvarez-Fallas, M. E., & Sancho, R. (2022). Differentiation of beta-like cells from human induced pluripotent stem cell-derived pancreatic progenitor organoids. *STAR Protocols*, 3(3). <https://doi.org/10.1016/J.XPRO.2022.101656>
- Piper, K., Brickwood, S., Turnpenny, L. W., Cameron, I. T., Ball, S. G., Wilson, D. I., & Hanley, N. A. (2004). Beta cell differentiation during early human pancreas development. *The Journal of Endocrinology*, 181(1), 11-23. <https://doi.org/10.1677/JOE.0.1810011>

- Russ, H. A., Parent, A. v, Ringler, J. J., Hennings, T. G., Nair, G. G., Shveygert, M., Guo, T., Puri, S., Haataja, L., Cirulli, V., Blelloch, R., Szot, G. L., Arvan, P., & Hebrok, M. (2015). Controlled induction of human pancreatic progenitors produces functional beta-like cells in vitro. *The EMBO Journal*, 34(13), 1759-1772. <https://doi.org/10.15252/EMBJ.201591058>
- Shih, H. P., Kopp, J. L., Sandhu, M., Dubois, C. L., Seymour, P. A., Grapin-Botton, A., & Sander, M. (2012). A Notch-dependent molecular circuitry initiates pancreatic endocrine and ductal cell differentiation. *Development (Cambridge)*, 139(14), 2488-2499. <https://doi.org/10.1242/DEV.078634/-/DC1>
- Trott, J., Tan, E. K., Ong, S., Titmarsh, D. M., Denil, S. L. I. J., Giam, M., Wong, C. K., Wang, J., Shboul, M., Eio, M., Cooper-White, J., Cool, S. M., Rancati, G., Stanton, L. W., Reversade, B., & Dunn, N. R. (2017). Long-Term Culture of Self-renewing Pancreatic Progenitors Derived from Human Pluripotent Stem Cells. *Stem Cell Reports*, 8(6), 1675. <https://doi.org/10.1016/J.STEMCR.2017.05.019>
- Wescott, M. P., Rovira, M., Reichert, M., von Burstin, J., Means, A., Leach, S. D., & Rustgi, A. K. (2009). Pancreatic Ductal Morphogenesis and the Pdx1 Homeodomain Transcription Factor. *Molecular Biology of the Cell*, 20(22), 4838. <https://doi.org/10.1091/MBC.E09-03-0203>

Rebuttal Figure 1. Lentiviral overexpression of USP7 in iPSC derived pancreas progenitors. a. Lentiviral constructs used for the generation of USP7 overexpressing PP. **b.** Bright field and GFP images from PP infected with pLV mCherry EGFP-T2A-Puro (Control) and pLV-Flag USP7 EGFP (USP7 overexpression). **c.** qPCR analysis for USP7 mRNA a week after the lentiviral infection (n=3 independent experiments; data presented as mean \pm SEM; ns ($p > 0.05$ after T-test analysis)).

REVIEWERS' COMMENTS

Reviewer #1 (Remarks to the Author):

The authors have done a great job to address my comments and improved the manuscript significantly. I believe that the current version of the manuscript is of high quality for publication. Only one minor point is that the authors need to check proper using of capital letters for showing gene/proteins in mouse and human in the text and the figures.

Reviewer #2 (Remarks to the Author):

In my previous review, I had raised a few concerns regarding the detailed information about the mechanism involved in the study and had requested the authors to clarify their rationale and conclusions. I am pleased to note that the authors have taken all the necessary steps to address my concerns and have provided a thorough response in the rebuttal letter. They have also combed through the entire manuscript and provided additional data from required experiments, which is commendable.

The authors have addressed my previous concerns, and I find the study to be intriguing as it provides insights into how Ngn3 (protein level) is regulated during endocrine development. The additional information and data provided by the authors have satisfied my requests for clarification, and I do not have any further requests for additional experiments. The additional analysis of the embryonic pancreas of Usp7 mutants further strengthened the study and supported the specificity (and time window) of Ngn3 regulation. The authors' additional experiment of inhibiting USP7 at different stages of stem cell differentiation using alternative protocols also provides further supporting evidence that USP7 acts on Ngn3 during the secondary transition. Although the gain of function for USP7 in iPSC may not be successful, in my opinion, the inhibition experiment sufficiently addresses the comments.

Overall, I believe that this study is suitable for publication in Nature Communications. The authors have demonstrated a clear understanding of the research and have taken the necessary steps to address concerns raised by reviewers. This study will be of great interest to the scientific community and will contribute significantly to our understanding of endocrine development. The authors should be commended for their hard work and dedication in producing such a high-quality study.

Response to reviewer comments for NCOMMS-22-20842-A 'USP7 controls NGN3 stability and pancreatic endocrine lineage development' by Manea et al.

We are pleased that the reviewers recognised the importance of our work on the post-translational regulation of Ngn3 by USP7, and the relevance for the field and that the reviewers agree that the study is now suitable for publication in Nature Communications. We are also pleased to see that the reviewers acknowledged our hard work and that we have demonstrated a clear understanding of the research and have taken the necessary steps to address concerns raised by reviewers. We would like to thank the reviewers for their critical assessment and constructive comments, which have greatly helped us to improve our study. We have now addressed one minor point from reviewer 1. The original referee comment is in *italic black text*, and our point-by-point response is in blue text below.

Reviewer #1 (Remarks to the Author):

The authors have done a great job to address my comments and improved the manuscript significantly. I believe that the current version of the manuscript is of high quality for publication. Only one minor point is that the authors need to check proper using of capital letters for showing gene/proteins in mouse and human in the text and the figures.

We have now checked each gene/proteins names within the text and figures and are now corrected in the revised version of our manuscript and figures.